# A highly conserved host lipase deacylates oxidized phospholipids and ameliorates acute lung injury in mice

Benkun Zou[1], Michael Goodwin[2], Danial Saleem[2], Wei Jiang[1], Jianguo Tang[3], Yiwei Chu[1], Robert S Munford[2], Mingfang Lu[1,3]*

[1]Department of Immunology, Key Laboratory of Medical Molecular Virology (MOE, NHC, CAMS), School of Basic Medical Sciences & Shanghai Institute of Infectious Disease and Biosecurity, Fudan University, Shanghai, China; [2]Laboratory of Clinical Immunology and Microbiology, NIAID, NIH, Bethesda, United States; [3]Department of Trauma-Emergency & Critical Care Medicine, Shanghai Fifth People's Hospital, Fudan University, Shanghai, China

**Abstract** Oxidized phospholipids have diverse biological activities, many of which can be pathological, yet how they are inactivated in vivo is not fully understood. Here, we present evidence that a highly conserved host lipase, acyloxyacyl hydrolase (AOAH), can play a significant role in reducing the pro-inflammatory activities of two prominent products of phospholipid oxidation, 1-palmitoyl-2-glutaryl-sn-glycero-3-phosphocholine and 1-palmitoyl-2-(5-oxovaleroyl)-sn-glycero-3-phosphocholine. AOAH removed the sn-2 and sn-1 acyl chains from both lipids and reduced their ability to induce macrophage inflammasome activation and cell death in vitro and acute lung injury in mice. In addition to transforming Gram-negative bacterial lipopolysaccharide from stimulus to inhibitor, its most studied activity, AOAH can inactivate these important danger-associated molecular pattern molecules and reduce tissue inflammation and injury.

*For correspondence: mingfanglu@fudan.edu.cn

Competing interest: The authors declare that no competing interests exist.

## Editor's evaluation

The article demonstrates that endogenous oxidized phospholipids (OxPLs) can stimulate inflammatory responses in host cells, including induction of cell death and processing of interleukin-1 β. The article further identifies a host enzyme, acyloxyacyl hydrolase (AOAH), that cleaves oxidized phospholipids and thereby reduces their pro-inflammatory properties. These are important findings that shed light on the mechanisms that limit damaging inflammation, and the findings should be of broad interest.

## Introduction

Oxidized phospholipids (oxPLs) are produced by peroxidation of cell membrane or lipoprotein phospholipids, usually during inflammation or senescence (*Bochkov et al., 2010*). They can be recognized by many receptors in the innate immune system, often with significant consequences (*Bochkov et al., 2017*; *Bochkov et al., 2010*; *Erridge et al., 2008*; *Hazen, 2008*; *Matt et al., 2015*; *Mauerhofer et al., 2016*). Some oxPLs may induce cell death or endothelial barrier disruption (*Fruhwirth and Hermetter, 2008*). For example, 1-palmitoyl-2-glutaroyl-sn-glycero-3-phosphocholine (PGPC) and 1-palmitoyl-2-(5-oxovaleroyl)-sn-glycero-3-phosphocholine (POVPC), two important oxidation products of 1-palmitoyl-2-arachidonoyl-sn-glycero-3-phosphocholine (PAPC), can activate acid sphingomyelinase to produce ceramide that then induces cell death of smooth muscle cells and macrophages

(*Greig et al., 2012*; *Loidl et al., 2003*; *Stemmer et al., 2012*). In endoplasmic reticulum-stressed macrophages, oxPLs may also trigger apoptosis in a CD36-TLR2 dependent manner (*Seimon et al., 2010*), and ozone-treated lung surfactant component 1-palmitoyl-2-(9′-oxo-nonanoyl)-sn-glycero-3-phosphocholine can induce macrophage death (*Uhlson et al., 2002*). OxPAPC components such as PGPC and POVPC may disrupt endothelial barriers in the lung while a full-length oxygenated product, 1-palmitoyl-2-(5,6-epoxyisoprostane E2)-sn-glycero-3-phosphatidyl choline (PEIPC), was found to protect the barrier (*Birukova et al., 2013*).

In addition to inducing cell death, oxPLs are able to evoke inflammatory responses in macrophages (*Bochkov et al., 2010*; *Vladykovskaya et al., 2011*). For example, PGPC and POVPC may induce IL-1β release from macrophages or dendritic cells that have been primed by lipopolysaccharide (LPS) or other TLR agonists in a NLRP3 and caspase 1/11-dependent manner (*Yeon et al., 2017*; *Zanoni et al., 2017*). Zanoni et al. have found that PGPC and POVPC, unlike other inflammasome activators, can induce IL-1β release without inducing pyroptosis (*Evavold et al., 2018*; *Hagar et al., 2013*; *Kayagaki et al., 2011*; *Kayagaki et al., 2013*; *Mangan et al., 2018*; *Shi et al., 2015*; *Shi et al., 2014*; *Zanoni et al., 2016*; *Zanoni et al., 2017*). In contrast, cyclo-epoxycyclopentenone, another oxPAPC component, can induce caspase 8-mediated IL-1β maturation and inflammatory apoptosis in primed macrophages or dendritic cells (*Muri et al., 2020*). Building on evidence that cell-surface CD14 can bind and internalize glycerophosphoinositides (*Wang et al., 1998*; *Wang and Munford, 1999*), Zanoni et al. found that CD14 captures and internalizes PGPC and POVPC and showed that this mode of cell entry is required for these oxPLs to induce macrophages to release IL-1β (*Zanoni et al., 2017*). Although macrophages play an essential role in many inflammatory responses, how they degrade oxPLs and the biological impact of this degradation are not well understood.

Acyloxyacyl hydrolase (AOAH), a highly conserved animal lipase, is expressed in phagocytes, including macrophages, microglia, dendritic cells and neutrophils, and also in NK cells, type 1 innate lymphoid cells, and renal proximal tubule cells (*Munford et al., 2020*). AOAH transforms bacterial LPS by cleaving the ester bonds that attach the secondary ('piggyback') fatty acyl chains to the primary (glucosamine-bound) hydroxyacyl chains in the lipid A moiety, converting a potent stimulus to a competitive inhibitor (*Munford et al., 2020*). In addition, AOAH can remove fatty acyl chains from the sn-1 and sn-2 positions of many glycero (phospho) lipids and be an acyl transferase (*Gorelik et al., 2018*; *Munford and Hunter, 1992*; *Staab et al., 1994*).

Here, we present evidence that AOAH can (a) reduce the ability of two prominent oxPLs (PGPC and POVPC) as well as an intermediate product of their deacylation (lysophosphatidycholine [LPC]) to induce cell death and IL-1β release from primed macrophages in vitro and (b) prevent oxPL-induced acute lung injury in vivo in mice.

## Results

### Low concentrations of oxPLs induce cell death and IL-1β release from primed macrophages

We began by asking if low concentrations of oxPAPC, PGPC, and POVPC can induce cell death and IL-1β release from mouse resident peritoneal macrophages. Cells were either not primed or primed with 10 ng/ml TLR2/1 agonist Pam3CSK4 (Pam3) for 7 hr and then a test lipid (PGPC, POVPC, oxPAPC, or non-oxidized PAPC) was added to the culture media for an additional 18 hr. We found that 5–12.5 µg/ml (8–21 µM) PGPC or POVPC and 12.5 µg/ml oxPAPC could induce significant cell death with or without Pam3 priming, whereas 12.5 µg/ml (16 µM) non-oxidized PAPC was inactive (*Figure 1A*). PGPC- and POVPC-induced IL-1β secretion occurred only in Pam3-primed cells and was dose-dependent (*Figure 1B*). Non-oxidized PAPC did not induce IL-1β release by primed cells (*Figure 1B*), in agreement with previous findings that only oxidized PLs activate the inflammasome (*Zanoni et al., 2016*; *Zanoni et al., 2017*). Pam3-induced IL-6 production was minimally affected by the presence of oxPLs (*Figure 1C*). Mature IL-1β (p17) appeared only after Pam3-primed macrophages had been treated with either PGPC or POVPC (*Figure 1D*). Previous studies have shown that PGPC or POVPC induced IL-1β release from primed bone marrow-derived macrophages (BMDMs) in a NLRP3-, caspase 1/11-dependent manner (*Yeon et al., 2017*; *Zanoni et al., 2017*), while Muri et al. have found that caspase 8 but not caspase 1 or NLRP3 was required for cyclo-epoxycyclopentenone-induced IL-1β processing (*Muri et al., 2020*). We found that caspase 1 inhibitor VX-765, caspase 8

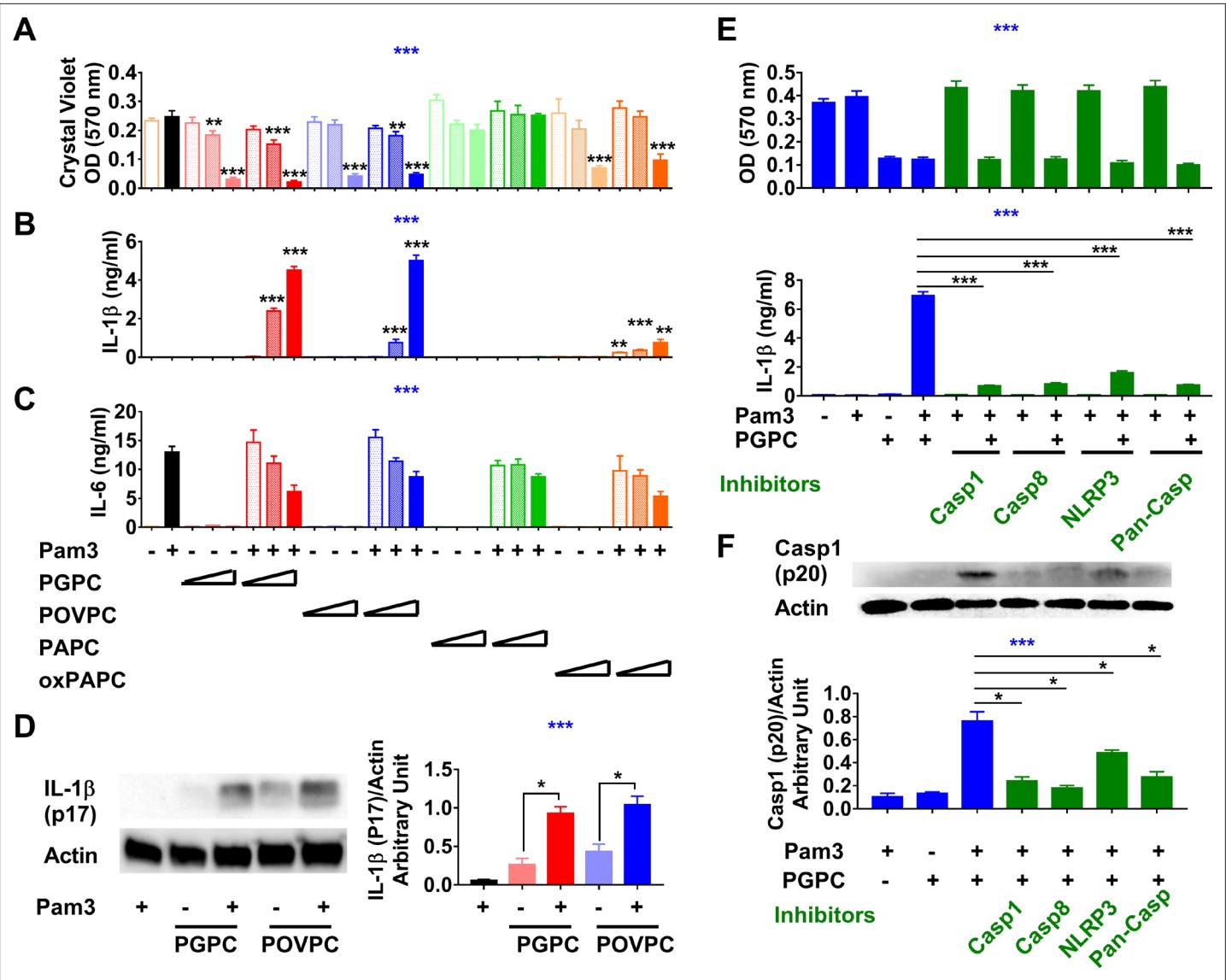

**Figure 1.** Low concentrations of oxidized phospholipids (oxPLs) induce macrophage cell death but IL-1β release requires priming. (**A–C**) Resident peritoneal macrophages were incubated for 7 hr with or without priming with 10 ng/ml Pam3. 2, 5, or 12.5 μg/ml PGPC, POVPC, PAPC, or oxPAPC were added to the media. After incubation for 18 hr, cells were washed and then stained with crystal violet (OD 570 nm). IL-1β and IL-6 were measured in the culture media using ELISA (**B, C**). Data were combined from at least two experiments, n ≥ 3. (**D**) Macrophages were treated with 10 ng/ml Pam3 for 7 hr and then 12.5 μg/ml PGPC or POVPC was added. After 18 hr incubation, the culture medium and cell lysate were combined, concentrated, and subjected to western blot analysis for mature IL-1β and actin. One representative experiment of four is shown (left). The western blot results were quantitated using ImageJ and expressed as IL-1β (p17)/actin (right). n = 4. (**E, F**) 10 μM VX-765 (caspase 1 inhibitor), 20 μM Z-IETD-FMK (caspase 8 inhibitor), 20 μM Z-VAD-FMK (Pan-Caspase inhibitor), or 100 μM MCC950 (NLRP3 inflammasome inhibitor) was added 1 hr before priming. After 7 hr priming with 10 ng/ml Pam3, macrophages were treated with 12.5 μg/ml PGPC for 18 hr. Cells were then stained with crystal violet and IL-1β in culture medium was measured (**E**). Data were combined from three experiments, each with n = 4. A mixture of culture medium and cell lysate was subjected to western blot analysis for caspase 1 (p20) and actin (**F**). One representative experiment of four is shown. The western blot results were quantitated using ImageJ and expressed as caspase 1 (p20)/actin, n = 4. One-way ANOVA was used to test the difference among groups (blue stars in **A–C**, **E, F**), and the Mann–Whitney test was used to test the difference between groups (black stars). In (**A–C**), the statistical tests were made between oxPL-treated cells and untreated cells with or without priming respectively. *p<0.05; **p<0.01; ***p<0.001.

The online version of this article includes the following figure supplement(s) for figure 1:

**Source data 1.**

**Source data 2.**

**Figure supplement 1.** Caspase 11 is not required for PGPC-induced IL-1β release in resident peritoneal macrophages.

*Figure 1 continued on next page*

*Figure 1 continued*

Figure supplement 1—source data 1.

Figure supplement 1—source data 2.

inhibitor Z-IETD-FMK, NLPR3 inhibitor MCC950, and pan-caspase inhibitor Z-VAD-FMK significantly reduced IL-1β release and caspase 1 processing in primed macrophages but did not prevent oxPL-induced cell death, demonstrating that oxPL-induced cell death does not require inflammasome activation (*Stemmer et al., 2012*; *Figure 1E and F*). The results suggest that caspase 8 and caspase 1 may act tandemly for IL-1β processing (*Chi et al., 2014*; *Philip et al., 2014*). Zanoni et al. have shown that oxPAPC-induced IL-1β release is dependent upon caspase 11 (*Zanoni et al., 2016*). In contrast, we found that caspase 11 is not required for PGPC-induced IL-1β release from peritoneal macrophages that had been primed (*Figure 1—figure supplement 1*). Low concentrations of oxPLs thus induced cell death; the inflammasome was activated and IL-1β was released if the macrophages had been primed.

## AOAH deacylates and inactivates oxPLs

AOAH can act in vitro as a phospholipase $A_1$, $A_2$, or B (*Gorelik et al., 2018*; *Munford and Hunter, 1992*). To find out if the enzyme can deacylate oxPLs, we incubated PGPC and POVPC with purified recombinant human AOAH (rhAOAH) for 2 or 4 hr. Analysis using LC-MS showed that AOAH could remove the sn-2 oxidized fatty acyl chain and sn-1 palmitate from both PGPC and POVPC (*Figure 2A and B*); the sn-2 moiety was removed preferentially, in keeping with the enzyme's preference for removing shorter acyl chains from glycerolipids (*Erwin and Munford, 1990*; *Gorelik et al., 2018*; *Munford and Hunter, 1992*). PGPC or POVPC treated with AOAH (dPGPC or dPOVPC) had greatly reduced ability to induce cell death (*Figure 2C and D*) and IL-1β release (*Figure 2E and F*) in Pam3-primed macrophages, while Pam3 stimulated similar levels of IL-6 production (*Figure 2E*) in the presence of either oxPLs or deacylated oxPLs. AOAH thus decreased the ability of these oxPLs to induce cell death and inflammasome activation.

## AOAH also deacylates and inactivates LPC

When AOAH released the sn-2 oxidized acyl chains from PGPC or POVPC, the major product was LPC (*Figure 2A and B*), another bioactive DAMP with disease associations (*Liu et al., 2020*); the appearance of free palmitate was evidence for AOAH-mediated deacylation of LPC (*Munford and Hunter, 1992*). When we incubated pure LPC with rhAOAH, we again found that palmitate was released (*Figure 3A*). In agreement with previous reports that LPC can be inflammatory and induce IL-1β release (*Li et al., 2016*; *Liu-Wu et al., 1998*; *Vladykovskaya et al., 2011*), we found that LPC induced cell death and IL-1β release by macrophages; as was observed with PGPC, LPC induced IL-1β release in a caspase 1, caspase 8, and NLRP3-dependent manner in primed macrophages, while inflammasome activation was not required for LPC-induced cell death (*Figure 3B and C*). AOAH treatment partially diminished these bioactivities, including a major reduction in IL-1β release (*Figure 3D*). In response to LPC, Pam3-primed *Aoah*[-/-] macrophages released more caspase 1 (p20) than did primed *Aoah*[+/+] macrophages (*Figure 3E*). Thus, AOAH can deacylate and inactivate LPC in vitro and prevent LPC-induced inflammasome activation.

## AOAH reduces oxPL-induced cell death and inflammasome activation in macrophages

We next explored a potential role for AOAH in inactivating oxPLs in cells by studying the responses of *Aoah*[+/+] and *Aoah*[-/-] resident peritoneal macrophages to PGPC and POVPC. After Pam3 priming, PGPC or POVPC treatment led to more cell death (*Figure 4A*) and more IL-1β release in *Aoah*[-/-] macrophages than in *Aoah*[+/+] controls (*Figure 4B*). Although AOAH can deacylate synthetic lipopeptides, including Pam3 (*Gorelik et al., 2018*), AOAH did not regulate IL-6 secretion in response to Pam3 stimulation with or without oxPLs (*Figure 4C*). Therefore, AOAH does not prevent macrophages from responding to Pam3, which can initiate cell signaling by engaging cell-surface TLR2/1, in keeping with previous findings that AOAH does not modulate acute responses to LPS, such as secretion of TNF-α, IL-6, or RANTES (*Lu et al., 2008*). In addition, we found that oxPLs induced significantly greater

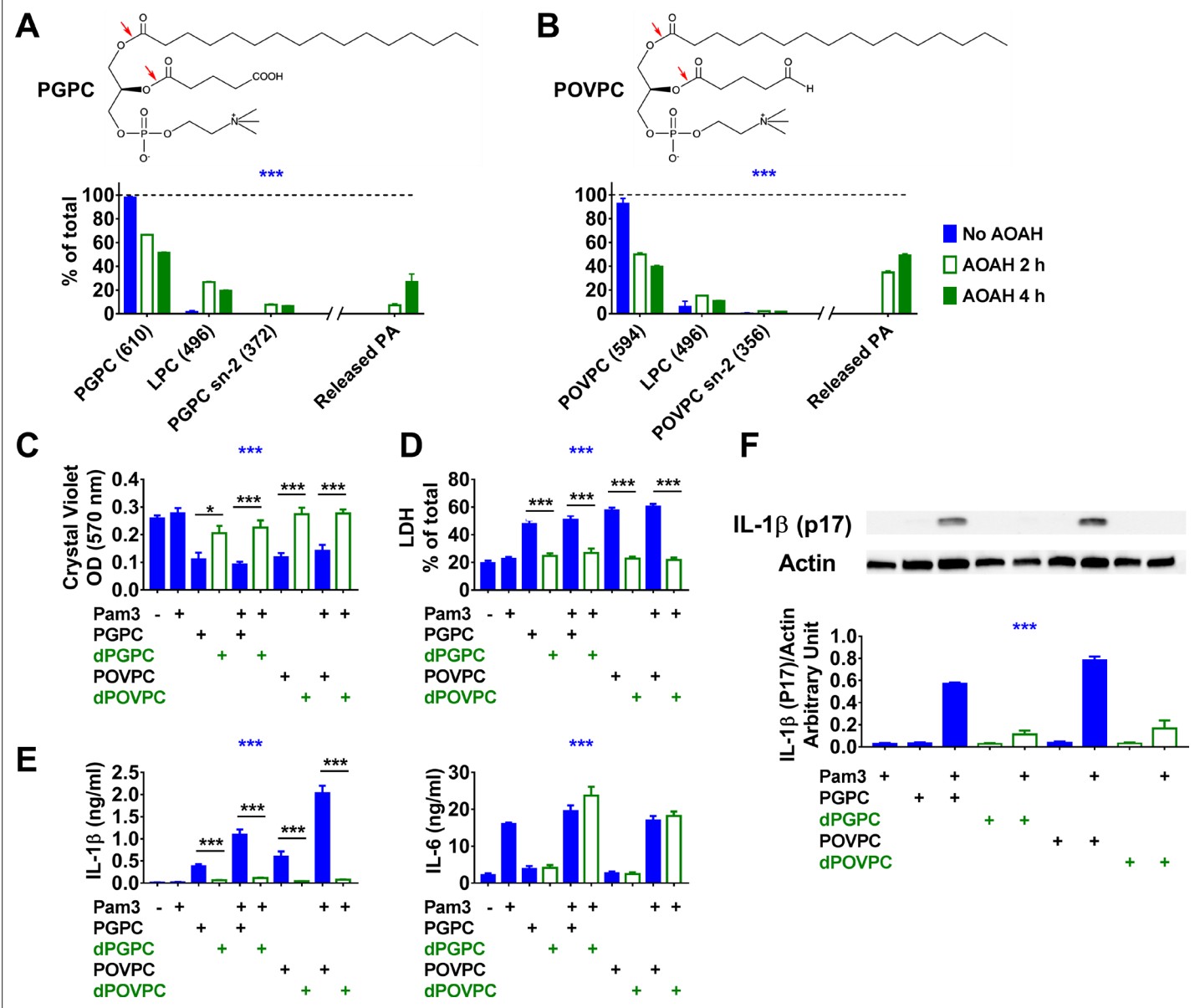

**Figure 2.** Acyloxyacyl hydrolase (AOAH) treatment decreases the bioactivities of PGPC and POVPC. (**A, B**) Chemical diagrams of PGPC and POVPC are shown. Red arrows indicate AOAH cleavage sites. PGPC and POVPC were incubated with rhAOAH at 37°C for 2 or 4 hr in 100 mM NaCl with 10 mM sodium acetate, pH 6.2, 0.1% Triton X-100, and 0.2 mg/ml fatty acid-free human BSA. The reaction products missing sn-2 oxidized FA (MW 496, LPC) or missing sn-1 palmitic acid (PA; MW 372 for PGPC or MW 356 for POVPC) were analyzed (see Materials and methods); the released PA was also measured. n = 4 (PGPC) or 3 (POVPC). (**C–F**) Resident peritoneal macrophages were treated or untreated with 10 ng/ml Pam3 for 7 hr. PGPC, POVPC, or AOAH-deacylated PGPC or POVPC (dPGPC or dPOVPC) (12.5 μg/ml) was then added. After incubation for 18 hr, cells were stained with crystal violet (OD 570 nm) and the OD was measured (**C**). Released LDH was measured (**D**). Concentrations of IL-1β and IL-6 (**E**) were measured in the culture media using ELISA. Data were combined from three experiments, each with n = 3 or 4 (**C–E**). Combined culture medium and cell lysate were subjected to western blot analysis and quantitated (**F**), n = 3. Two-way ANOVA was used to test the difference among groups (no AOAH, AOAH 2 and 4 hr, blue stars in **A, B**). One-way ANOVA was used to test the difference among groups (blue stars in **C–F**), and the Mann–Whitney test was used to test the difference between groups (black stars). *p<0.05; **p<0.01; ***p<0.001.

The online version of this article includes the following figure supplement(s) for figure 2:

**Source data 1.**

**Source data 2.**

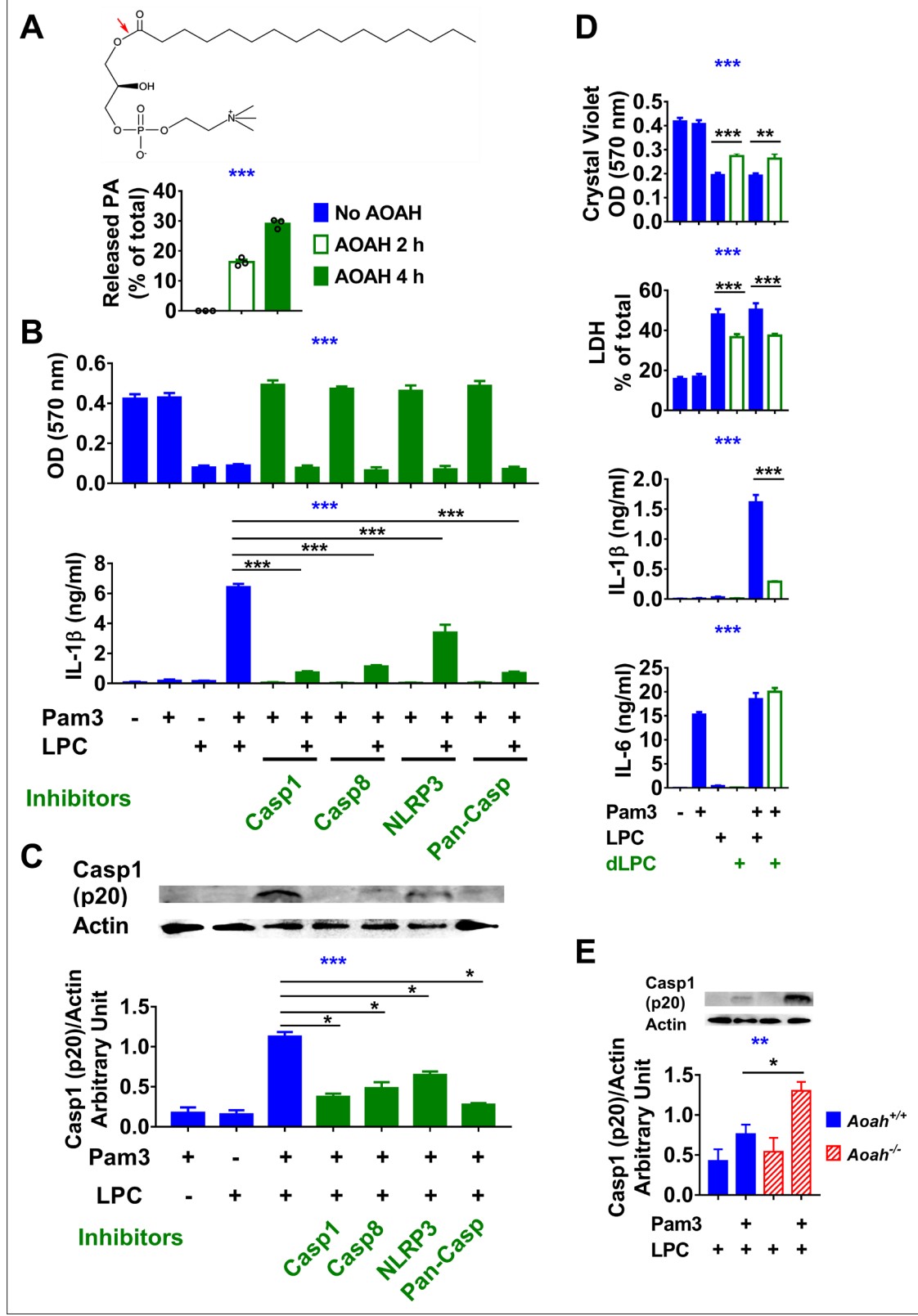

**Figure 3.** Acyloxyacyl hydrolase (AOAH) treatment decreases the bioactivities of lysophosphatidycholine (LPC). (**A**) The chemical diagram of LPC is shown. The red arrow indicates the AOAH cleavage site. After LPC was incubated with purified rhAOAH for 2 or 4 hr, the released palmitate was measured. No palmitate was detected when rhAOAH was absent. n = 3. (**B, C**) Resident peritoneal macrophages were treated with 10 μM VX-765, 20 μM Z-IETD-FMK, 20 μM Z-VAD-FMK, or 100 μM MCC950 for 1 hr, and then primed with 10 ng/ml Pam3 for 7 hr before 12.5 μg/ml (25 μM) LPC was

*Figure 3 continued on next page*

*Figure 3 continued*

added to the media. After 18 hr incubation, cells were stained with crystal violet (OD 570 nm) and medium IL-1β was measured (**B**). Data were combined from three experiments, each with n = 3. Cleaved caspase 1 (p20) in the combined medium and cell lysate was analyzed by western blot, and the blots were quantitated using ImageJ, n = 4 (**C**). (**D**) Macrophages were treated or untreated with 10 ng/ml Pam3 for 7 hr before 12.5 µg/ml (25 µM) LPC or AOAH-treated LPC (deacylated LPC, dLPC) was added. After incubation for 18 hr, cells were stained with crystal violet and the medium was used to measure LDH activity, IL-1β and IL-6 concentrations. Data were combined from two (LDH) or four (crystal violet, IL-1β and IL-6) experiments, each with n = 4. (**E**) *Aoah*[+/+] and *Aoah*[-/-] macrophages were treated with 10 ng/ml Pam3 for 7 hr and then 5 µg/ml LPC was added. After 18 hr incubation, the media were subjected to western blot analysis for cleaved caspase 1 (p20), and the cell lysates were used for actin. Western blot results from four experiments were quantitated using ImageJ. One-way ANOVA was used to test the difference among groups (blue stars in **A–E**), and the Mann–Whitney test was used to test the difference between groups (black stars). *p<0.05; **p<0.01; ***p<0.001.

The online version of this article includes the following figure supplement(s) for figure 3:

**Source data 1.**

**Source data 2.**

caspase 1 processing in Pam3-primed *Aoah*[-/-] macrophages, confirming that AOAH diminishes oxPL-induced inflammasome activation (*Figure 4D*). To test the possibility that AOAH might also decrease the potency of the priming agent, we primed cells with CpG DNA, a TLR9 agonist that is not an AOAH substrate. After CpG-primed macrophages were treated with PGPC, more cell death and IL-1β release were again found in *Aoah*[-/-] macrophages than in *Aoah*[+/+] controls (*Figure 4E*), confirming that AOAH regulates the bioactivities of oxPLs. When oxPLs are added to culture media, CD14-dependent internalization of oxPLs is required for inflammasome activation (*Zanoni et al., 2017*). When we delivered PGPC or POVPC into the cell cytosol using N-[1-(2,3-dioleoyloxy)propyl]-N,N,N-trimethylammonium methyl-sulfate (DOTAP), bypassing CD14 and endosomes (*Zanoni et al., 2017*), primed *Aoah*[+/+] and *Aoah*[-/-] macrophages released similar amounts of IL-1β (*Figure 4F*) and cleaved caspase 1 (p20) (*Figure 4G*), in keeping with previous findings that AOAH localizes in acidic cellular vesicles instead of cytosol (*Luchi and Munford, 1993*; *Munford and Hunter, 1992*; *Staab et al., 1994*). In addition, AOAH did not alter cytosolic LPS-induced IL-1β release (*Figure 4H*; *Shi et al., 2014*; *Zanoni et al., 2016*) or modulate ATP-induced inflammasome activation or IL-1β release in Pam3-primed macrophages (*Figure 4I*), evidence that AOAH does not prevent inflammasome activation per se. These results suggest that oxPL internalization via CD14-induced endocytosis may allow AOAH, which is found in endolysosomes (*Luchi and Munford, 1993*; *Staab et al., 1994*), to deacylate oxPLs before they activate the inflammasome.

## AOAH regulates inflammatory responses and lung injury induced by oxPLs

Having found in vitro that (a) rhAOAH can deacylate PGPC, POVPC, and LPC and reduce their ability to stimulate macrophages and (b) AOAH produced by resident macrophages can reduce oxPL-induced cell death and inflammasome activation, we next used four acute lung injury models to study the enzyme's role in vivo. We first tested whether oxPLs can induce AOAH expression in alveolar macrophages (AMs). We found that, unlike LPS, which stimulated AOAH mRNA expression in AMs by 100-fold in vivo (*Zou et al., 2017*), oxPLs induced AOAH expression by only seven-fold (*Figure 5A*). Moreover, oxPLs reduced LPS-induced *Aoah* expression by AMs, suggesting that oxPLs may act as partial agonists to induce AOAH expression (*Figure 5A*).

As we had observed that sequential and simultaneous treatment of macrophages with LPS and oxPL induced similar IL-1β production (*Figure 5—figure supplement 1*), we instilled *Aoah*[+/+] and *Aoah*[-/-] mice i.n. with 40 µl PBS containing 10 µg LPS, oxPLs (25 µg PGPC and 25 µg POVPC), or LPS plus oxPLs. Whereas *Aoah*[+/+] and *Aoah*[-/-] mice had similar levels of protein in their bronchoalveolar lavage fluid (BALF) (reflecting alveolar leakage) 18 hr after either LPS or the oxPLs were introduced, *Aoah*[-/-] mice that received both LPS and oxPLs had more BALF protein and more neutrophils in their BALF at this time than did identically treated *Aoah*[+/+] mice (*Figure 5B and C*). Notably, after oxPLs were instilled, *Aoah*[-/-] mice had more monocytes-macrophages in their airspaces than did *Aoah*[+/+] mice (*Figure 5C*), suggesting that AOAH may regulate oxPL-induced inflammation. After LPS and oxPL instillation, both inflammatory (IL-6, TNF-α, IL-1β, MCP-1) and anti-inflammatory (IL-10) cytokine mRNA were more abundant in *Aoah*[-/-] mouse lungs (*Figure 5D*). The mRNA levels of IRAK-M, a negative regulator of TLR signaling pathway, were also elevated in *Aoah*[-/-] mouse lungs (*Figure 5D*).

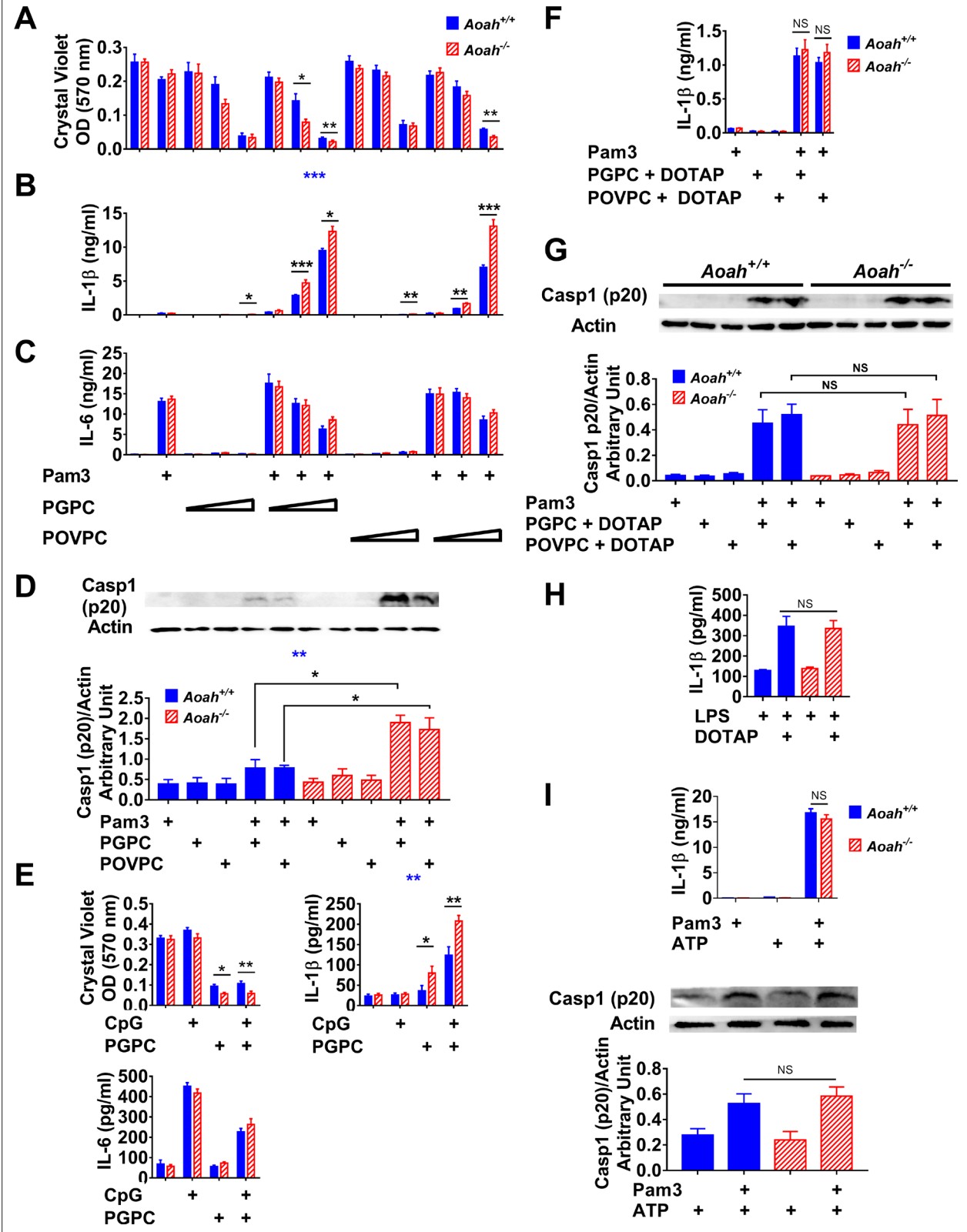

**Figure 4.** Acyloxyacyl hydrolase (AOAH) reduces oxidized phospholipid (oxPL)-induced cell death and inflammasome activation in macrophages. (**A–C**) *Aoah*[+/+] and *Aoah*[-/-] resident peritoneal macrophages were incubated for 7 hr with or without priming with 10 ng/ml Pam3. 2, 5, or 12.5 µg/ml PGPC or POVPC were then added to the media. After incubation for 18 hr, cells were washed and then stained with crystal violet (**A**). IL-1β (**B**) and IL-6 (**C**) were measured in the culture media using ELISA. Data were combined from three experiments, each with n = 4 wells/group. (**D**) Macrophages were primed

*Figure 4 continued on next page*

*Figure 4 continued*

with 10 ng/ml Pam3 and then 5 µg/ml PGPC or POVPC was added. After 18 hr incubation, the culture media were concentrated and subjected to western blot analysis for cleaved caspase 1 (p20). Cell lysate was used for actin detection. Western blot results from four experiments were quantitated using ImageJ and plotted below the western images, n = 4. (**E**) Macrophages were primed with 1 µM CpG DNA for 7 hr and then 12.5 µg/ml PGPC was added. After 18 hr incubation, the cells were washed and stained with crystal violet and the media were collected for IL-1β and IL-6 measurement using ELISA. Data were combined from three experiments, each with n = 3–4. (**F, G**) Macrophages were primed with 10 ng/ml Pam3 and then 5 µg/ml PGPC or POVPC encapsulated in DOTAP was added. After 18 hr incubation, the media were collected for IL-1β detection using ELISA (**F**) and subjected to western blot analysis for cleaved caspase 1 (p20). Most cells remained alive after the treatment. Cell lysates were collected for actin detection (**G**). Data were combined from 2 (F, n = 3 in each experiment) or four experiments (**G**). (**H**) Macrophages were treated with 1 µg/ml lipopolysaccharide (LPS) for 21 hr or primed with 1 µg/ml LPS for 3 hr and then 1 µg/ml LPS mixed with DOTAP was added to deliver LPS into the cytosol to activate caspase 11. After 18 hr incubation, IL-1β was measured in the culture medium. Data were combined from two experiments, each with n = 5. (**I**) Macrophages were primed with 10 ng/ml Pam3 for 7 hr and then 2 mM ATP was added to induce inflammasome activation. After 18 hr incubation, the culture medium was analyzed using ELISA. Data were combined from two experiments, each with n = 3/group. The mixture of medium and cell lysate was used for western blot analysis, and the results from three experiments were quantitated using ImageJ. n = 4/group. Two-way ANOVA was used to test the difference between *Aoah^{+/+}* and *Aoah^{-/-}* groups with multiple treatments (blue stars), and the Mann–Whitney test was used to test the difference between groups that received the same treatment (black stars). *p<0.05; **p<0.01; ***p<0.001.

The online version of this article includes the following figure supplement(s) for figure 4:

**Source data 1.**

**Source data 2.**

The BALF from *Aoah^{-/-}* mouse contained more IL-6, TNF-α, and IL-1β than that from *Aoah^{+/+}* mice (**Figure 5E**). These results suggest that AOAH may reduce oxPL-induced lung inflammation and injury. As AOAH can also inactivate LPS, to examine more specifically the enzyme's role in inactivating oxPLs we also used heat-inactivated Gram-positive bacteria, *Streptococcus pneumoniae,* to induce inflammation in the lung. We instilled heat-killed *S. pneumoniae* i.n., with or without oxPLs, to *Aoah^{+/+}* and *Aoah^{-/-}* mice. We did not observe significant differences in pulmonary inflammation between the two strains of mice when they received only *S. pneumoniae* instillation (**Figure 5E and F**). Following instillation of *S. pneumoniae* plus oxPLs, we found that *Aoah^{-/-}* mice had greater leukocyte infiltration in their airspaces and more pulmonary IL-6, MCP-1, and KC mRNA than did *Aoah^{+/+}* mice, providing further evidence that AOAH can act on oxPLs to ameliorate lung inflammation (**Figure 5F and G**).

## AOAH reduces bioactive oxPLs in the lungs after HCl and oxPL instillation

In a third animal model, we used HCl and oxPLs to induce pulmonary inflammation in *Aoah^{+/+}*, *Aoah^{-/-}*, and transgenic mice that express high levels of AOAH in their macrophages (*Ojogun et al., 2009*). When we instilled only HCl and studied the mice 18 hr later, *Aoah^{-/-}* and *Aoah^{+/+}* mice had weak and similar alveolar barrier damage (**Figure 6A**) and pulmonary inflammation (**Figure 6B–D**). When both HCl and oxPLs were instilled, in contrast, *Aoah^{-/-}* mice had much more alveolar barrier damage and lung inflammation than did *Aoah^{+/+}* mice (**Figure 6A–D**). In addition, the transgenic mice that express high levels of AOAH (*Ojogun et al., 2009*) had reduced airway inflammation after instillation of HCl plus oxPLs (**Figure 6—figure supplement 1**). These findings again support a significant role for AOAH in modulating oxPL-induced pulmonary injury.

To obtain further evidence that AOAH modulates oxPL bioactivity in vivo, we used E06, a natural IgM that binds to the hydrophilic head of oxidized PCs such as POVPC and blocks their ability to induce inflammation in vitro and in vivo (*Friedman et al., 2002*; *Que et al., 2018*; *Shaw et al., 2000*; *Sun et al., 2020*). We detected more E06-binding activity in BALF from *Aoah^{-/-}* mice, suggesting that *Aoah^{-/-}* mice were less able to degrade oxPLs in their airways (**Figure 6E**). We used BALF to stimulate Pam3-primed macrophages and found that BALF from HCl- and oxPL-instilled *Aoah^{-/-}* mice induced significantly more IL-1β production than did BALF from identically treated *Aoah^{+/+}* mice (**Figure 6F**). Pre-incubation with E06 Ab diminished the ability of BALF from *Aoah^{-/-}* mice to induce IL-1β production (**Figure 6F**). Taken together, these results again support the conclusion that AOAH can degrade and inactivate oxPLs in vivo.

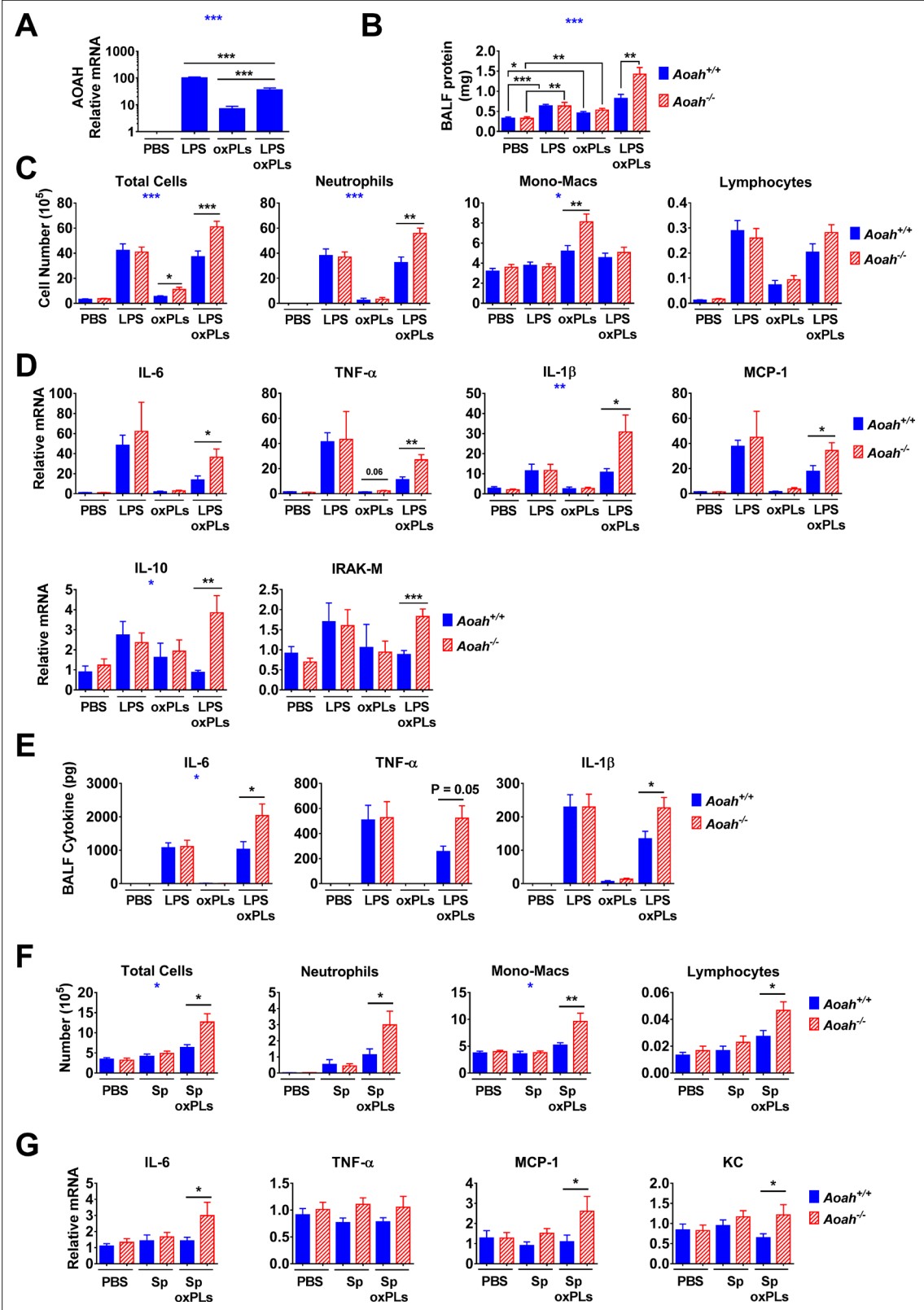

**Figure 5.** Acyloxyacyl hydrolase (AOAH) reduces inflammatory responses and lung injury induced by oxidized phospholipids (oxPLs). (A–E) *Aoah*[+/+] mice were instilled i.n. with 10 µg lipopolysaccharide (LPS), oxPLs (25 µg PGPC + 25 µg POVPC), or LPS + oxPLs. 18 hr later, AOAH expression in alveolar macrophages (AMs) was measured using qPCR. Data were combined from three experiments, n = 9 mice/group (A). Bronchoalveolar lavage fluid (BALF) protein amount was measured (B). BAL immune cells were analyzed using cytospin followed by Wright–Giemsa staining (C). Lung cytokine/

*Figure 5 continued on next page*

*Figure 5 continued*

chemokine and IRAK-M mRNA were measured using qPCR (**D**). BALF cytokines were measured using ELISA (**E**). Data were combined from two (**E**) or three (**B–D**) experiments, each with n = 3 mice/group. (**F, G**) Mice were instilled with 40 µl PBS containing 6 × 10 exp[6] heat-inactivated *Streptococcus pneumoniae*, with or without oxPLs (25 µg PGPC and 25 µg POVPC). 18 hr later, their BALF cells were analyzed (**F**). Lung cytokine/chemokine expression was measured using qPCR (**G**). Data were combined from at least two experiments, each with n = 3 mice/group. One-way ANOVA was used to test the difference among groups (**A**, blue stars), two-way ANOVA was used to test the difference between *Aoah*[+/+] and *Aoah*[-/-] groups with multiple treatments (**B–G**, blue stars), and the Mann–Whitney test was used to test the difference between groups that received the same treatment (black stars). *p<0.05; **p<0.01; ***p<0.001.

The online version of this article includes the following figure supplement(s) for figure 5:

**Source data 1.**

**Figure supplement 1.** Sequential and simultaneous treatment of macrophages with lipopolysaccharide (LPS) and PGPC induced similar IL-1β production.

**Figure supplement 1—source data 1.**

## AOAH regulates inflammatory responses induced by endogenous oxPLs

In the previous HCl-induced lung inflammation models, we instilled exogenous oxPLs mixed with HCl i.n. (*Figure 6*). To test whether AOAH can regulate endogenous oxPL-induced lung inflammation, we used an acid aspiration-induced acute lung injury model that has been shown to induce oxPLs in the lung (*Imai et al., 2008*). Mice were instilled with 40 µl PBS or 0.2 M HCl, ventilated for 1 hr, and allowed to recover off the ventilator for 24 hr before analysis. The HCl-instilled and ventilated *Aoah*[-/-] mice had elevated BALF protein, which was similar to the damage induced by HCl plus oxPLs but less than that induced by LPS and oxPLs (*Figure 7A*). The ventilated *Aoah*[-/-] mice had more neutrophils and IL-6 in their airspaces as well as higher levels of lung inflammation than did *Aoah*[+/+] control mice (*Figure 7B–D*), especially when the mice had been instilled with HCl. Histological study showed that HCl-instilled, ventilated *Aoah*[-/-] mouse lungs had more leukocyte infiltration and alveolar barrier thickening than did control mouse lungs (*Figure 7E*). BALF from *Aoah*[-/-] mice not only bound more E06 antibody than did BALF from *Aoah*[+/+] mice (*Figure 7F*), but it also had greater ability to induce IL-1β release from Pam3-primed macrophages (*Figure 7G*). This activity was partially inhibited by E06 (*Figure 7G*). These results provide additional evidence that AOAH is able to inactivate oxPLs generated de novo in vivo.

## Discussion

Acting as danger-associated molecular patterns (DAMPs), oxPLs may contribute to the pathogenesis of acute lung injury, atherosclerosis, non-alcoholic fatty liver disease, neurodegenerative disorders, and cancer (*Bochkov et al., 2017*; *Bochkov et al., 2010*; *Lee et al., 2012*). Previous studies have described several mechanisms for inactivating these phospholipids. Plasma platelet-activating factor acetyl hydrolase (PAF-AH) and intracellular PAF-AH II, which cleave sn-2 oxidized acyl chains from PGPC and POVPC, are GDSL lipases that have structural similarity to AOAH; lecithin-cholesterol-acyltransferase transfers oxidized acyl chains from oxPLs to cholesterol (*Goyal et al., 1997*); and oxidized acyl chains can be reduced by enzymes such as aldose reductase, phospholipid glutathione peroxidase, peroxiredoxin 6, and glutathione transferase (*Mauerhofer et al., 2016*; *Vladykovskaya et al., 2011*). Phospholipase $A_2$ was implicated in POVPC deacylation in human THP-1 cells, which produce very little AOAH; the resulting LPC was more stimulatory than was POVPC itself (*Vladykovskaya et al., 2011*). Autophagy also helps remove oxPLs (*Bochkov et al., 2010*; *Mauerhofer et al., 2016*).

AOAH was originally found to cleave the ester bonds that attach the piggyback (secondary) fatty acyl chains to the primary 3-hydroxy acyl chains in the lipid A moiety of LPS (*Hall and Munford, 1983*; *Munford and Hall, 1986*). AOAH-deacylated (tetraacyl) LPS has greatly reduced stimulatory potency and can inhibit LPS signaling by competing with LPS for binding to LPS-binding protein, soluble and membrane CD14, and MD-2 (*Erwin and Munford, 1990*; *Kitchens et al., 1992*; *Munford and Hall, 1986*; *Munford and Hall, 1989*). Subsequent studies have found that AOAH plays a significant role in inactivating both exogenously administered LPS and LPS derived from intestinal commensal

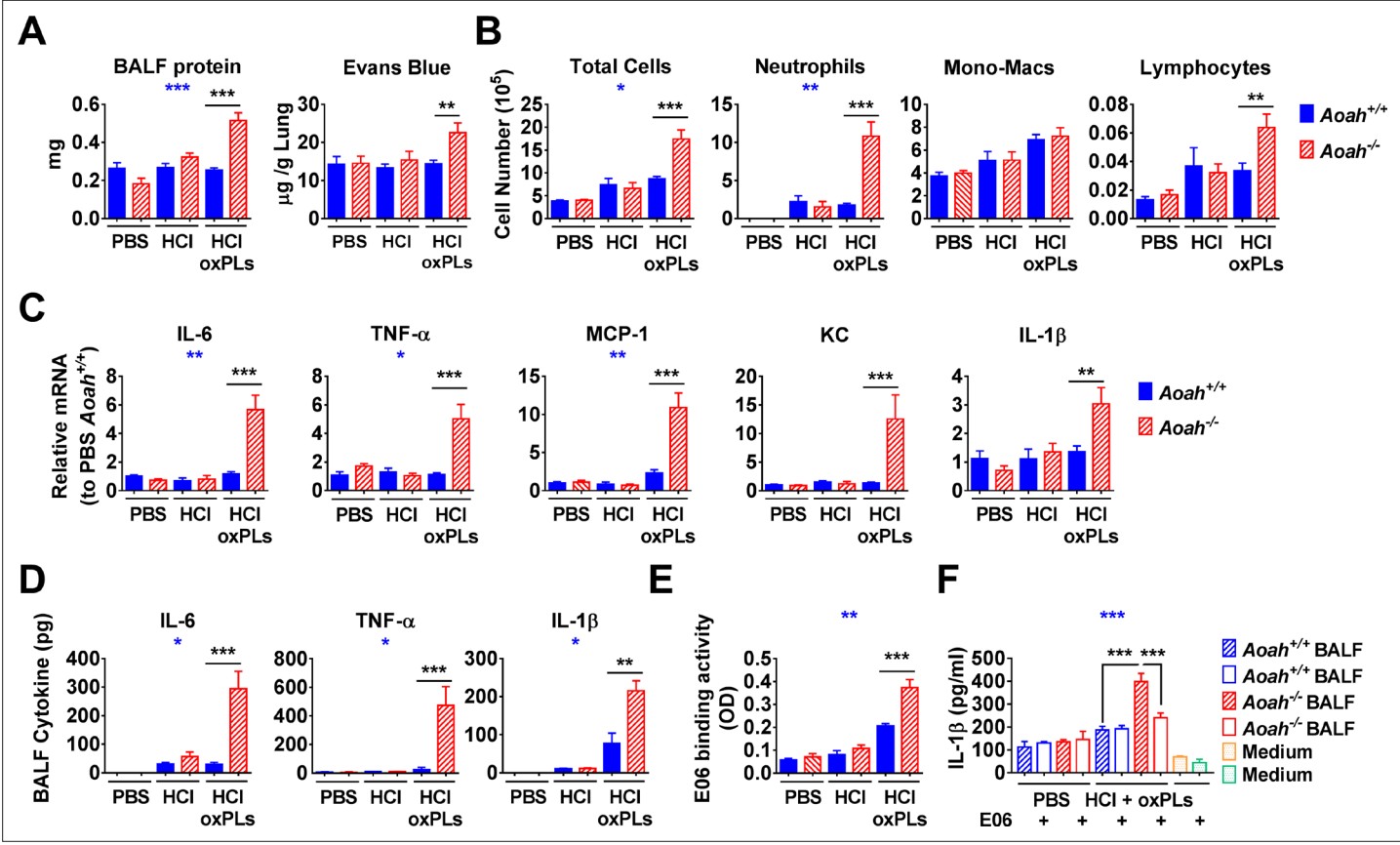

**Figure 6.** Acyloxyacyl hydrolase (AOAH) reduces bioactive oxidized phospholipids (oxPLs) in the airspaces after HCl and oxPL instillation. (**A**) *Aoah*[+/+] and *Aoah*[-/-] mice were instilled with 40 µl 0.2 M HCl and oxPLs (25 µg PGPC + 25 µg POVPC). 18 hr later, bronchoalveolar lavage fluid (BALF) was obtained. Protein in BALF was quantitated. 17 hr after i.n. HCl and oxPL instillation, Evans blue was injected i.v. 1 hr after injection, the lung extravascular dye was extracted and measured. Data were combined from two or three experiments, each with n = 3 mice/group. (**B–F**) In the same experiments as in (**A**), cells in BALF were analyzed after cytospin and Wright–Giemsa staining (**B**). Lung cytokine/chemokine expression was measured by qPCR (**C**). Data were combined from at least two experiments, each with n = 3 mice/group (**B, C**). Cytokines in BALF were analyzed using ELISA (**D**). E06-detectable oxPLs in BALF were measured (**E**). BALF was used to stimulate Pam3-primed macrophages for 18 hr in the presence of 5 µg/ml E06 or a control mouse IgM. IL-1β in the media was measured using ELISA (**F**). (**D–F**) Data were combined from two experiments, n = 4–12 mice/group. Two-way ANOVA was used to test the difference between *Aoah*[+/+] and *Aoah*[-/-] groups with multiple treatments (blue stars), and the Mann–Whitney test was used to test the difference between two groups that received the same treatment (black stars). *p<0.05; **p<0.01; ***p<0.001.

The online version of this article includes the following figure supplement(s) for figure 6:

**Source data 1.**

**Figure supplement 1.** Mice that overexpress acyloxyacyl hydrolase (AOAH) in macrophages have reduced airway inflammation after HCl and oxidized phospholipid (oxPL) challenge.

**Figure supplement 1—source data 1.**

Gram-negative bacteria (*Han et al., 2021*; *Lu et al., 2013*; *Lu et al., 2008*; *Lu et al., 2005*; *Qian et al., 2018*; *Shao et al., 2011*; *Shao et al., 2007*; *Zou et al., 2017*). AOAH was also found to have phospholipase, lysophospholipase, diacylglycerollipase, and acyltransferase activities in vitro (*Gorelik et al., 2018*; *Munford and Hunter, 1992*; *Staab et al., 1994*). In the present study, we found that AOAH can deacylate both oxidized glycerophospholipids and LPC and diminish their ability to induce macrophage cell death and stimulate inflammasome activation in vitro. Using four mouse models, we found that AOAH diminishes oxPL-induced acute lung injury.

In previous studies, we found that AOAH does not prevent acute responses to LPS. Instead, the enzyme modulates the late, or longer-lasting, responses to LPS stimulation (*Lu et al., 2008*; *Lu et al., 2005*; *Shao et al., 2007*; *Zou et al., 2017*). *Aoah*[+/+] and *Aoah*[-/-] mice had similar degrees of lung injury 24 hr after LPS was instilled i.n., while on day 4, *Aoah*[-/-] mice showed delayed resolution of pulmonary inflammation (*Zou et al., 2017*). *Aoah*[+/+] and *Aoah*[-/-] resident peritoneal macrophages had comparable

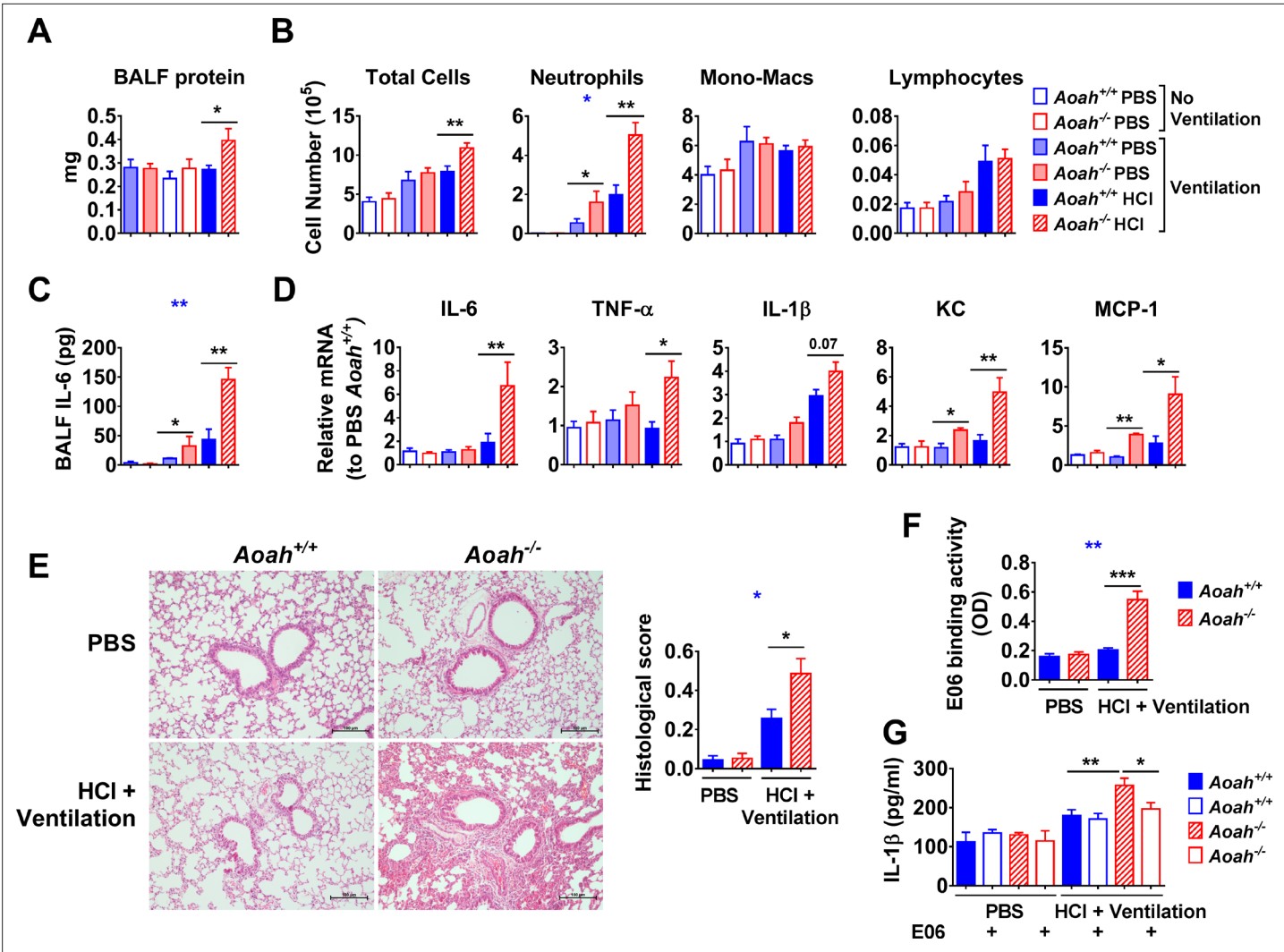

**Figure 7.** Acyloxyacyl hydrolase (AOAH) regulates inflammatory responses induced by endogenous oxidized phospholipids (oxPLs). *Aoah⁺/⁺* and *Aoah⁻/⁻* mice were instilled with 40 µl PBS or 0.2 M HCl, and then they were ventilated for 1 hr. 24 hr later, the mice were analyzed. (**A**) Bronchoalveolar lavage fluid (BALF) protein amount was measured. n = 4 or 5. (**B**) After cytospin and Wright–Giemsa staining, immune cells in BALF were counted. (**C**) IL-6 in BALF was measured using ELISA. TNF-α and IL-1β were undetectable. (**D**) Cytokine/chemokine expression was measured in the lungs. Data (**B–D**) were combined from at least two experiments, each with n = 3 mice/group. (**E**) Mouse lungs were excised and fixed in paraformaldehyde. The fixed lungs were sectioned and stained with hematoxylin-eosin. One representative picture of eight mice in each group is shown. The lung samples were scored according to the lung injury scoring system recommended by the American Thoracic Society, n = 8 mice/group. (**F**) BALF was collected and E06 binding activity was measured. (**G**) BALF was used to stimulate Pam3-primed macrophages, and the IL-1β in culture medium was measured. Data were combined from at least two experiments, each with n = 4 mice/group (**F, G**). Two-way ANOVA was used to test the difference between *Aoah⁺/⁺* and *Aoah⁻/⁻* groups with multiple treatments (blue stars), and the Mann–Whitney test was used to test the difference between groups that received the same treatment (black stars). *p<0.05; **p<0.01; ***p<0.001.

The online version of this article includes the following figure supplement(s) for figure 7:

**Source data 1.**

**Source data 2.**

innate responses to TLR stimuli (*Lu et al., 2013*; *Lu et al., 2008*). In contrast, AOAH efficiently limits inflammasome activation by oxPLs. This may be related in part to its location in endolysosomes (*Luchi and Munford, 1993*; *Staab et al., 1994*), where oxPLs may be delivered when they are internalized by CD14 (*Kagan et al., 2008*; *Zanoni et al., 2011*; *Zanoni et al., 2017*), and by the enzyme's ability to inactivate both oxPLs and LPCs. By inactivating oxPLs in endolysosomes, AOAH prevents oxPL leakage into the cytosol and endolysosomal damage, diminishing inflammasome activation (*Figure 8*).

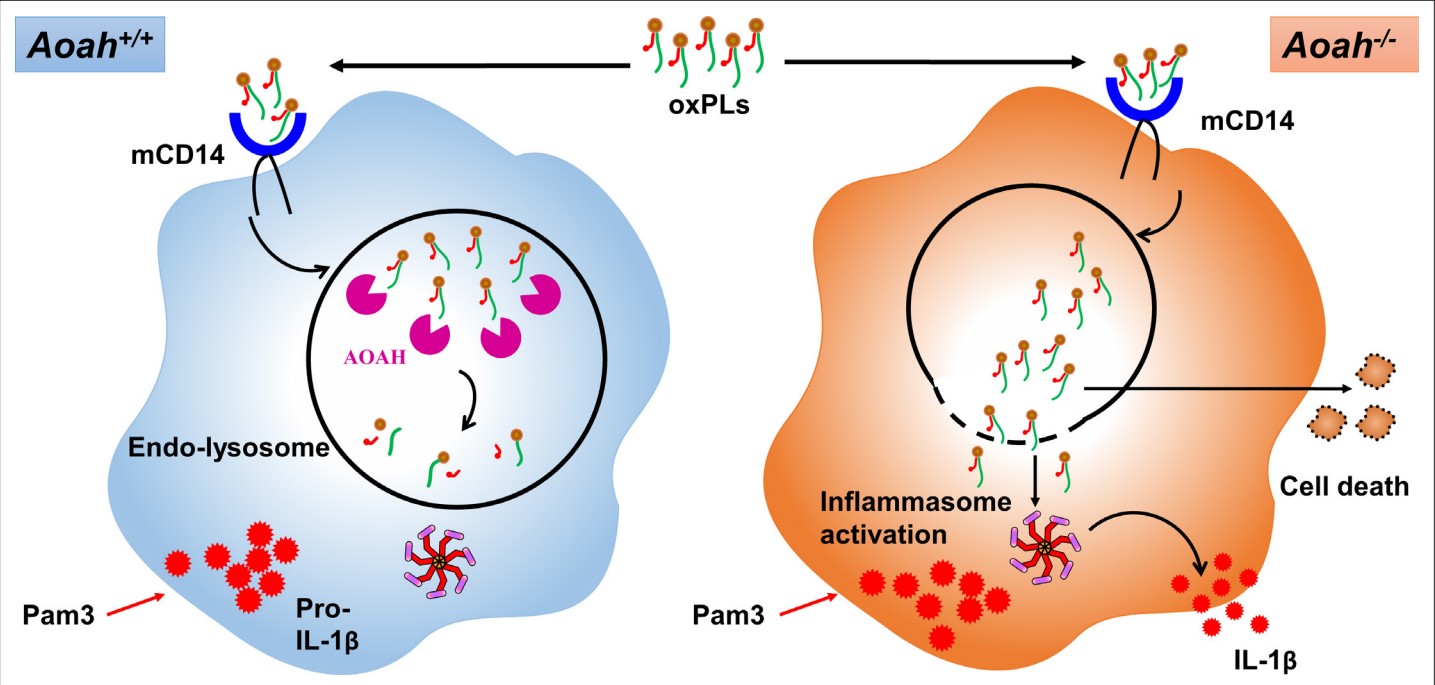

**Figure 8.** Acyloxyacyl hydrolase (AOAH) deacylates and inactivates oxidized phospholipids (oxPLs). After oxPLs enter endolysosomes in a CD14-dependent pathway, AOAH in endolysosomes deacylates and inactivates oxPLs. When AOAH is lacking, oxPLs leak into cytosol or endolysosome membrane ruptures, leading to exaggerated inflammasome activation and IL-1β release.

Thus, while endolysosomal AOAH cannot deacylate LPS before it is sensed by TLR4, the enzyme may act on oxPLs before they activate the inflammasome.

Previous studies have shown that oxPLs may have both pro-inflammatory and anti-inflammatory bioactivities (*Bochkov et al., 2002*; *Erridge et al., 2008*). Low concentrations of oxPLs have usually induced protective effects while high concentrations have been stimulatory/toxic (*Mauerhofer et al., 2016*; *Muri et al., 2020*; *Oskolkova et al., 2010*). Zanoni et al. found that PGPC or POVPC activated the inflammasome in primed dendritic cells or macrophages (*Zanoni et al., 2016*; *Zanoni et al., 2017*), and we confirmed their results with lower concentrations of oxPLs. In contrast, Chu et al. found that a low concentration (2 µg/ml) of oxPAPC directly bound caspase 11 and inhibited LPS-induced pyroptosis, IL-1β release, and septic shock (*Chu et al., 2018*). The discrepant effects reflect different experimental conditions. When Chu et al. either transfected LPS and oxPAPC simultaneously into the cytosol of macrophages or treated macrophages with oxPAPC first and then transfected LPS, they found reduced cell death and IL-1β release. In this experimental setting, oxPAPC, as a partial agonist, may compete with the strong agonist, LPS. In our study, we primed macrophages with microbe-associated molecular patterns (MAMPs) such as LPS or Pam3 for several hours to increase inflammasome component expression and then added oxPLs to activate the inflammasome. In our experiments, LPS did not enter the cytosol to activate caspase 11. Even when we added LPS and PGPC/POVPC simultaneously to macrophages, we also observed inflammasome activation, suggesting that macrophages may be primed by TLR4 stimulation before oxPLs are taken up via CD14 and then transferred from endolysosomes to cytosol to activate the inflammasome (*Zanoni et al., 2017*). In addition, we used specific oxPAPC components, PGPC or POVPC, instead of oxPAPC with its many components and diverse bioactivities. Thus, the oxPL species used and their concentrations, the routes of MAMP and oxPL delivery, and the order of addition of MAMP and oxPLs may contribute to the observations made in different laboratories. In this study, we focused on defining the role that AOAH plays in preventing oxPL-induced inflammasome activation. Whether AOAH also plays a role in regulating or promoting the anti-inflammatory activities of oxPLs awaits further investigation.

In summary, we have presented in vitro and in vivo evidence that a host lipase, AOAH, can deacylate and inactivate two important oxPLs and also LPC. It has been intriguing that CD14 can transfer both LPS and oxPLs into endosomes (*Kitchens and Munford, 1998*; *Zanoni et al., 2017*) and that

TLR4 and caspase 11 can recognize both oxPLs and LPS (*Hagar et al., 2013*; *Imai et al., 2008*; *Kayagaki et al., 2011*; *Kayagaki et al., 2013*; *Shi et al., 2014*; *Zanoni et al., 2016*); here we show that the same lipase can inactivate both of these agonists and prevent them from inducing inflammasome activation. As oxPLs and lysophospholipids can contribute to many inflammatory diseases and aging, inactivating these lipids may be a very important function of this highly conserved enzyme.

## Materials and methods

### Prime and stimulate macrophages

Many experimental variables may influence the impact that oxPLs have on cells in vitro, and both pro- and anti-inflammatory responses have been reported (*Bochkov et al., 2002*; *Chu et al., 2018*; *Erridge et al., 2008*; *Mauerhofer et al., 2016*; *Muri et al., 2020*; *Oskolkova et al., 2010*). Influential variables have included the oxPLs tested and their concentrations (e.g., below or above the lipid's critical micellar concentration [CMC]), the incubation times, the production of inflammatory LPC from oxPL by intracellular lipases (*Vladykovskaya et al., 2011*), the culture media serum concentration (*Stemmer et al., 2012*), 'priming' vs. co-incubation, and the method used to introduce the oxPL (or co-stimulus) into cells (transfection into cytosol vs. internalization via CD14 into endolysosomes) (*Chu et al., 2018*; *Zanoni et al., 2017*). The cells studied are also important as some (macrophages and dendritic cells) produce AOAH and others (THP-1 cells, human endothelial cells) do not.

For these experiments, we used sub-CMC concentrations of two known agonists, PGPC and POVPC, long incubation times to allow oxPL uptake via CD14 and catabolism by AOAH, and 0.1% serum to minimize binding of oxPL to serum proteins (*Erridge et al., 2008*) and extracellular oxPL deacylation (*Stemmer et al., 2012*).

Peritoneal cells were harvested from naïve mice and cultured in RPMI medium (Gibco) containing 10% fetal bovine serum (FBS, Gibco), 2 mM glutamine, 100 U/ml penicillin, and 0.1 mg/ml streptomycin (Life Technologies). After incubation for 4 hr at 37°C to let the macrophages adhere to culture plates, the floating cells were washed away and the peritoneal macrophages were incubated in RPMI medium containing 0.1% FBS, 100 U/ml penicillin, and 0.1 mg/ml streptomycin with or without 10 ng/ml Pam3CSK4 (InvivoGen) or 10 ng/ml LPS O111:B4 (Sigma) for 7 hr to prime macrophages. PGPC (Cayman), POVPC (Cayman), PAPC (Sigma), oxPAPC (oxidized PAPC, Avanti), or 1-palmitoyl-2-hydroxy-sn-glycero-3-phosphocholine (lysophosphatidycholine or LPC, Cayman) in ethanol were dried under nitrogen and then suspended in RPMI containing 0.1% FBS with vigorous vortexing, water bath sonication, and further vortexing. The lipids were then added to culture medium to stimulate primed or unprimed macrophages for 18 hr. We used phospholipid concentrations that were below their CMCs, which are 54 µM, 68 µM, and 50 µM for PGPC, POVPC, and LPC, respectively, and similar to levels found in animal tissues (*Bochkov et al., 2010*; *Fruhwirth and Hermetter, 2008*; *Mauerhofer et al., 2016*; *Pande et al., 2010*). Brief incubation in 10% serum, used for many in vitro studies of oxPL activities, has been reported to convert POVPC and PGPC to LPC (*Stemmer et al., 2012*). We used 0.1% serum to avoid this issue.

We used DOTAP (Roche) to introduce LPS or oxPLs into cells following the method by *Zanoni et al., 2016*. Briefly, 375 ng DOTAP was used to encapsulate 5 µg LPS or 10 µg oxPLs in 10 µl Opi-MEM (Thermo). After 30 min incubation at 37°C, the DOTAP/LPS or DOTAP/oxPLs were diluted and added to cell culture media at final concentrations of 1 µg/ml or 5 µg/ml, respectively.

### Cell death analysis

Crystal violet (CV) (Sigma) was used to measure adherent cell DNA as described (*Feoktistova et al., 2016*). LDH cytotoxicity assay (Pierce) was done according to the manufacturer's instruction.

### Western blot and ELISA

After macrophages were primed with Pam3 and challenged with 5 µg/ml PGPC, POVPC, or LPC, >80% of cells were alive. Culture medium was collected, medium protein was precipitated using the chloroform-methanol method described by *Fic et al., 2010*. The protein pellet was dissolved in SDS loading buffer containing 10% β-mercaptoethanol and then subjected for western blot of caspase 1 (p20). The cell lysate was used for actin detection. When macrophages were treated with 12.5 µg/ml PGPC or POVPC, ~80% cells died after 18 hr treatment; the culture medium and cells were combined,

concentrated, and subjected to western blot analysis of IL-1β (p17), caspase 1 (p20), and actin. All of the concentrated proteins from each well were used for western blot. IL-1β (p17), caspase 1 (p20), and actin were detected using monoclonal antibodies D4T2D, E2G2I (Cell Signaling Technology), and 2D4H5 (Proteintech), respectively. The western blots were quantitated using ImageJ (NIH). Mouse IL-1β (R&D Systems), TNF-α, and IL-6 ELISAs (BD) were performed according to the manufacturer's instructions.

## AOAH activity toward PGPC, POVPC, and LPC in vitro

PGPC and POVPC (Cayman) (100 nmoles) were dried under argon before 5 µl ethanol was added to facilitate resuspension. Fatty acid-free BSA in 0.9% saline (0.2 mg/ml final concentration) and Triton X-100 (0.09%) were added while vortexing under argon, and the suspension was briefly sonicated in a water bath. 10 mM sodium acetate was added to adjust pH to 5.9–6.1. After mixing thoroughly and removing 50 µl (no AOAH control), 1 µl of 1 mg/ml recombinant human acyloxyacyl hydrolase (rhAOAH, provided by ZymoGenetics and stored at –80°C [*Hagen et al., 1991*]) was added to 450 µl reaction mixture, 50 µl of the mixture were aliquoted into polypropylene microfuge tubes, and incubation was carried out at 37°C with intermittent rotation for 2 or 4 hr. 10 µl were removed from each tube to measure C8 or longer free fatty acids (i.e., palmitic acid released from PGPC or POVPC) (Sigma free fatty acid kit MAK044), and the remaining samples were stored at –80°C prior to use or analysis. Freezing in the absence of glycerol inactivates AOAH.

The AOAH-treated PGPC and POVPC reaction products were analyzed using an Agilent 1290 Infinity UPLC coupled to a 6460C Agilent triple quadrupole mass selective detector (LC-QqQ). Samples were separated on an EclipsePlus C18 column 2.1 mm × 50 mm 1.8 mm particle, using aq. 0.1% formic acid (mobile phase A) and acetonitrile with 0.1% formic acid (mobile phase B). Flow rate was 0.8 ml/min beginning with 5% B holding 0.25 min, ramping to 95% B over 9 min, hold 4 min, and re-equilibrate. Mass spectrometer was operated in multiple reaction monitoring (MRM) mode with M+H$^+$ ion as the precursor and product ion m/z 184.1 (phosphatidyl choline group) created using collision energy voltage (CEV) 24–28 V. Fractions were calculated using % signal abundance observed in MRM. 'MW' in *Figure 2* legend refers to the monoisotopic mass + adduct (H$^+$).

To treat LPC, 1 µg rhAOAH was added to 450 nmoles LPC as above, and the pH was adjusted with 10 mM Na acetate to pH 5.9–6.1. After 2 or 4 hr incubation at 37°C, the released palmitate was measured using the Sigma free fatty acid assay kit. No free fatty acid was detected when AOAH was not added.

## Caspase inhibitors

10 µM VX-765 (caspase 1 inhibitor, Selleck), 20 µM Z-VAD-FMK (pan-caspase inhibitor, Selleck), 20 µM Z-IETD-FMK (casp-8 inhibitor, Selleck), or 100 µM MCC950 (NLRP3 inflammasome inhibitor, Selleck) was added to the culture medium for 1 hr before priming and stimulation as described previously.

## Animal models

Imai et al. found that HCl/ventilation and influenza virus infection induces oxPLs in the lung (*Imai et al., 2008*). We used HCl/ventilation model and three other lung injury models with clinical relevance: LPS-induced, Gram-positive bacterial infection (*S. pneumoniae*)-induced, acid gastroesophageal reflux (HCl), and acid aspiration-ventilation-induced injury. Mice were randomly assigned to different treatment or control groups.

## Three intranasal instillation acute lung injury models

50 µg OxPLs (25 µg PGPC + 25 µg POVPC) in ethanol were added to a polypropylene tube, and the ethanol was evaporated using nitrogen. (a) 10 µg LPS, (b) 6 × 10$^6$ heat-inactivated *S. pneumoniae* bacteria suspended in 40 µl PBS, or (c) 40 µl 0.2 M HCl was used to resuspend dried 50 µg oxPLs. *S. pneumoniae bacteria* (*Sp*) were cultured on Columbia blood agar plates, and a single colony was picked and inoculated in 3% TSB (Tryptic Soy Broth) medium containing 10% FBS. After shaking at 150 rpm, 37°C for 10 hr, bacteria were collected by centrifugation, resuspended in PBS, and then heat-inactivated at 100°C for 10 min. The bacterial culture with 6.6 × 10$^8$ *Sp*/ml contained 0.012 EU/ml endotoxin, determined using the Endotoxin Detection kit (InvivoGen). Each of the suspensions then underwent vortexing for 30 s, water bath sonication for 10 min, and vortexing for another 30 s before

it was instilled intranasally into mice that had been anesthetized i.p. with 0.5% pentobarbital sodium (50 µg/g body weight). After 18 hr, mice were exsanguinated by cutting the inferior vena cava. Bronchoalveolar lavage (BAL) was performed by cannulating the trachea with a 20-gauge catheter that was firmly fixed with a suture. The lung was then infused with 1 ml PBS containing 5 mM EDTA, and BALF was harvested. This procedure was repeated five times, and the BALF was combined. The lung was then cut into pieces for preservation in RNA Later (TianGen) for RT-PCR or in 4% paraformaldehyde for histological analysis.

In some experiments, we used transgenic mice that express AOAH from the CD68 promoter in macrophages and dendritic cells (*Ojogun et al., 2009*).

## Acid aspiration-ventilation acute lung injury model

Mice were anesthetized i.p. with 1% pentobarbital sodium (100 µg/g body weight) to reach a state of deep anesthesia. An infrared heater was used to maintain body temperature. 40 µl 0.2 M HCl was instilled intratracheally using a 1 mm diameter capillary tube. A 20-gauge catheter was then inserted into the airway and connected to a mechanical respirator (Harvard apparatus). Respiration was carried out for 60 min with 100 breaths/min, tidal volume 0.5 ml, sigh volume 0.5 ml, sigh frequency 1/0, and inspiratory to expiratory ratio 1:2. The mice were then removed from the respirator and monitored. 24 hr later, mice were euthanized and analyzed.

## Analysis of BALF

After the lungs were lavaged with 1 ml PBS twice, BALF was collected and centrifuged. The cell-free supernatant was used to measure protein concentrations with a bicinchoninic acid (BCA) kit (Pierce). The total BALF protein amount equals protein concentration times 1 ml. Inflammatory cytokines and chemokines in BALF were measured using TNF-α, IL-6 (BD), or IL-1β (R&D Systems) ELISA kits. The cell pellet was resuspended in PBS, and total cell numbers were counted using Cellometer (Nexelcom). Cell differential counting was conducted using cytospin and Wright–Giemsa staining. After counting 300 cells, the proportions of neutrophils, monocytes-macrophages, and lymphocytes were calculated and used to determine the number of each cell type. To assess alveolar barrier damage, extravascular Evans blue was measured as described previously (*Zou et al., 2017*). Briefly, Evans blue was injected i.v. 1 hr before euthanasia. After perfusion of the lungs, extravascular Evans blue was extracted and the optical density was measured at 620 nm (reference 740 nm) using a Tecan reader.

## Real-time PCR

RNA from AMs or lungs was isolated using RNA isolation kit (Tiangen) and reversely transcribed (Tiangen). The following primers were used at a concentration of 10 µM: AOAH (5′-CAGCTACTCCCA TGGCCAAA-3′, 3′-GCCACCTGGACTGAAGAGTT-5′), actin (5′-GGCTGTATTCCCCTCCATCG-3′, 3′-CCAGTTGGTAACAATGCCATGT-5′), IL-6 (5′-ATCGTGGAAATGAGAAAAGAGTTGT-3′, 3′-AAGTGCAT CATCGTTGTTCATACA-5′), TNF-α (5′-CATCTTCTCAAAATTCGAGTGACAA-3′, 3′-TCAGCCACTCCA GCTGCTC-5′), MCP-1 (5′-TCCCAATGAGTAGGCTGGAG-3′, 3′-TCTGGACCCATTCCTTCTTG-5′), IL-1β (5′-CCTTCCAGGATGAGGACATGA-3′, 3′-GTCACACACCAGCA GGTTATCA-5′), KC (5′-CAAGAACA TCCAGAGCTTGAAGGT-3′, 3′-GTGGCTATGACTTCGGTTTGG-5′), IL-10 (5′-GCTGGACAACATACTG CTAACC-3′, 3′-ATTTCCGATAAGGCTTGGCAA-5′), and IRAK-M (5′-TCCCACCTGAGGTGAAGCAT-3′, 3′-TGTGACATTGGCTGGTTCCA-5′). Actin was used as the internal control, and relative gene expression was calculated by the ΔΔCt quantification method.

## Histology

Lungs were excised and fixed in 4% paraformaldehyde containing 5% sucrose and 2 mM EDTA for 24 hr. The lung tissues were sectioned and stained with hematoxylin and eosin (H&E). The samples were examined using a Nikon E200 microscope and scored blinded based on the presence of neutrophils in the alveolar space or in the interstitial space, hyaline membranes, proteinaceous debris filling the airspaces and alveolar septal thickening (*Matute-Bello et al., 2011*). Ten random 400× fields were scored and averaged for each mouse.

### Detection of BALF-oxidized PCs by E06 Ab

A published method was used with modification (*Imai et al., 2008*). Briefly, after BALF was harvested and centrifuged, 100 µl supernatant was transferred to a high binding polystyrene plate (Corning) and incubated at 4°C for 18 hr. The plate was washed three times with PBS and blocked with 5% BSA in Tris-buffered saline pH 7 for 1 hr. 5 µg/ml E06 antibody (Avanti Polar Lipids) or a control mouse IgM (clone, MM-30, BioLegend) was added and incubated at room temperature for 2 hr. After washing the plate three times with PBS, anti-mouse IgM (H + L)-HRP antibody (Thermo) in 100 µl PBS was added and incubated at room temperature for 1 hr. The wells were washed seven times with PBS and TMB substrate (BD) was added (100 µl/well). After 30 min incubation at room temperature, the plate was read at 450 nm (reference 570 nm) (Tecan). OxPL-binding activity (OD) = OD with E06 – OD with the control IgM.

### OxPL neutralization by E06 Ab

In the lung inflammation models, the lungs were lavaged with 1 ml PBS twice and the BALF was centrifuged. The supernatant was transferred into a new tube and then incubated with 5 µg/ml E06 or a control mouse IgM for 30 min at 37°C. Pam3-primed macrophages were incubated with the BALF at 37°C for 18 hr and medium IL-1β was measured.

### Statistical analysis

Independent biological replicates are presented as mean ± SEM. One-way ANOVA was used to test the difference among groups, two-way ANOVA was used to test the difference between $Aoah^{+/+}$ and $Aoah^{-/-}$ groups with multiple treatments, and the Mann–Whitney test was used to test the difference between two groups that received the same treatment (two-tailed, unpaired). $*p<0.05$; $**p<0.01$; $***p<0.001$.

### Acknowledgements

This study was supported by grants 31770993, 91742104, and 31570910 from the National Natural Science Foundation of China (ML), grant 21ZR1405400 from the Shanghai Committee of Science and Technology (ML), and by the Divisions of Intramural Research, National Institute for Allergy and Infectious Diseases, NIH, USA (RSM, DS, MG). We thank Feng Shao for providing $Caspase11^{-/-}$ mice.

## Additional information

### Funding

| Funder | Grant reference number | Author |
|---|---|---|
| National Natural Science Foundation of China | 31770993 | Mingfang Lu |
| National Natural Science Foundation of China | 91742104 | Mingfang Lu |
| National Natural Science Foundation of China | 31570910 | Mingfang Lu |
| Shanghai Science and Technology Committee | 21ZR1405400 | Mingfang Lu |
| National Institute of Allergy and Infectious Diseases | The Divisions of Intramural Research | Robert S Munford Michael Goodwin Danial Saleem |

The funders had no role in study design, data collection and interpretation, or the decision to submit the work for publication.

### Author contributions

Benkun Zou, Formal analysis, Formal analysis, Investigation, Methodology, Validation, Writing – review and editing, Writing – original draft; Michael Goodwin, Formal analysis, Funding acquisition,

Investigation, Methodology, Validation, Writing – original draft; Danial Saleem, Wei Jiang, Investigation, Methodology; Jianguo Tang, Yiwei Chu, Methodology, Resources; Robert S Munford, Conceptualization, Formal analysis, Funding acquisition, Investigation, Methodology, Project administration, Resources, Supervision, Validation, Writing – original draft, Writing – review and editing; Mingfang Lu, Conceptualization, Formal analysis, Funding acquisition, Project administration, Resources, Supervision, Validation, Writing – original draft, Writing – review and editing

**Author ORCIDs**
Benkun Zou http://orcid.org/0000-0002-6980-0909
Danial Saleem http://orcid.org/0000-0003-2310-6062
Robert S Munford http://orcid.org/0000-0003-1509-1294
Mingfang Lu http://orcid.org/0000-0002-8612-3444

**Ethics**
All mice were studied using protocols approved by the Institutional Animal Care and Use Committee (IACUC) of Fudan University (20160824) and the National Institute of Allergy and Infectious Diseases (LCID-11e). All protocols adhered to the Guide for the Care and Use of Laboratory Animals. All surgery was performed under sodium pentobarbital anesthesia, and every effort was made to minimize suffering.

**Decision letter and Author response**
Decision letter https://doi.org/10.7554/eLife.70938.sa1
Author response https://doi.org/10.7554/eLife.70938.sa2

## Additional files

**Supplementary files**
• Transparent reporting form

**Data availability**
All data generated or analysed during this study are included in the manuscript and supporting files. Source data files have been provided for Figures 1—7 and Figure 1—figure supplement 1, Figure 5—figure supplement 1 and Figure 6—figure supplement 1.

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
