## [Editor Report]

The article demonstrates that endogenous oxidized phospholipids (OxPLs) can stimulate inflammatory responses in host cells, including induction of cell death and processing of interleukin-1 β. The article further identifies a host enzyme, acyloxyacyl hydrolase (AOAH), that cleaves oxidized phospholipids and thereby reduces their pro-inflammatory properties. These are important findings that shed light on the mechanisms that limit damaging inflammation, and the findings should be of broad interest.

---

## [Decision Letter]

**Decision letter after peer review:**

Thank you for submitting your article "A highly conserved lipase deacylates oxidized phospholipids and ameliorates acute lung injury" for consideration by *eLife*. Your article has been reviewed by 3 peer reviewers, one of whom is a member of our Board of Reviewing Editors, and the evaluation has been overseen by Paul Noble as the Senior Editor. The reviewers have opted to remain anonymous.

Essential revisions:

The manuscript demonstrates that endogenous oxidized phospholipids (OxPLs) can stimulate inflammatory responses in host cells, including induction of cell death and processing of interleukin-1 β. The manuscript futher identifies a host enzyme, acyloxyacyl hydrolase (AOAH), that cleaves oxidized phospholipids and thereby reduces their pro-inflammatory properties. A combination of in vitro and in vivo experiments supports the anti-inflammatory effects of AOAH. These are important findings that shed light on the mechanisms that limit damaging inflammation and we expect the findings to be of broad interest. The reviewers agreed that some technical aspects of the experiments and/or their presentation could be improved. We encourage submission of a revised version that address the following essential revisions:

(1) One of the main claims of the manuscript is that OxPLs have pro-inflammatory effects that include inflammasome activation (independent of Casp11). This claim is somewhat at odds with at least some of the existing literature which claims that OxPLs inhibit Casp11 inflammasome activation and/or synergize with LPS to produce enhanced Casp11-dependent IL-1 release. The apparent discrepancies with the literature are not adequately discussed or addressed.

(a) We require more discussion of how the new claims of this manuscript are to be reconciled with previous claims in the literature (e.g., see papers cited by R2).

(b) To support the main claim of inflammasome activation by OxPLs, we require that the authors address which inflammasome is being activated. At a minimum, we require that they test for the possible involvement of NLRP3 either with NLRP3 KO cells or MCC950 inhibitor.

(2) We require that technical issues with many of the experiments be addressed:

(a) All immunoblots require loading controls (see comments from all reviewers).

(b) Figure 2C-F: a better description of the experiment is required (see comments from R2).

(c) The blot in 3F needs quantification; if the effect is not significant, more caution is warranted.

(d) Clarify if AOAH KO cells have a growth defect, and if so, normalize the data in Figure 4 as requested by R2.

(e) Address the possible confounding issues associated with using Pam3CSK4 as a priming agent (see comments from R1).

(f) Address the issues raised about ZVAD/VX765 (see R1) and modify conclusions as necessary.

(g) Address the statistical issues identified by R3.

(3) The in vivo experiments using the HCl ventilation model (Figure 7) are particularly central to the main conclusion of the paper, namely, the claim that endogenous OxPLs are counteracted by AOAH (The other in vivo experiments involve provision of exogenous inflammatory lipids). As such, the data provided with this model in Figure 7 should be solidified:

(a) Improve the analysis of the histological findings in Figure 7 as suggested by R3.

(b) Quantify additional inflammatory markers, e.g., quantification of cytokines at the protein level, and markers of lung damage, e.g., quantification of Evans Blue and/or BAL protein as suggested by R3.

*Reviewer #1 (Recommendations for the authors):*

(1) It is difficult to know how to reconcile this work with other work in the field. It has been claimed that OxPLs antagonize Casp11 and are thus anti-inflammatory, whereas the current work suggests they are pro-inflammatory independent of Casp11. A more straightforward discussion of previous work and any relevant differences is necessary to help a reader put the present work in context.

(2) If Casp11 is not activated by OxPLs, which inflammasome is? At a minimum a simple experiment with NLRP3 KO cells (or MCC950) could rule in or out NLRP3, which is the most obvious candidate.

(3) Although it is appreciated that priming is necessary for this system, it is potentially confouding that many experiments use a lipid (Pam3CSK4) to prime since AOAH may potentially be acting on this lipid (and this may in part explain some of the observed effects). Can the authors address this possibility a bit more cleanly? Why not prime with a cytokine (e.g., TNF?) that would not itself be a possible substrate of AOAH?

(4) The experiments with ZVAD in S1 lack a ZVAD only control. In my experience ZVAD itself is highly toxic to macropahges so this control seems critical. In addition, since there is a CASP8 pathway downstream of inflammasome activation, CASP1 inhibitors are not sufficient to reach the conclusion that cell death is inflammasome-independent (e.g., p.6).

(5) Many of the western blots lack loading controls.

*Reviewer #2 (Recommendations for the authors):*

In this manuscript Zou et al., investigate the role of the lipase AOAH on the pro-inflammatory roles of oxidized lipids (PGPC, POVPC). They show that the removal of sn-1 and sn-2 acyl chains from these lipids by AOAH dampens oxPL induced inflammasome activation, in vitro and in vivo. They further demonstrate that AOAH also dampens inflammasome activation evoqued by LPC. In conclusion, AOAH reduces tissue inflammation and cell death in response to endogenous oxidized lipids.

Oxidized lipids have diverse biological activities, among these the induction of pathology and inflammation. Thus, the identification of an endogenous regulator that can dampen these effects is novel and interesting. The manuscript is understandable and the experiments can be easily followed. However some important controls are missing, and some point require clarification. Thus, the manuscript would improve if the following points would be addressed:

The OxPL field is highly controversial. For example, OxPAPAC has also been shown to block non-canonical inflammasome activation (PMID: 29520027), and in general acts as an anti-inflammatory molecule (PMID: 32234476). The authors show discuss these discrepancies and ideally provide experimental data to show how AOAH affect the anti-inflammatory functions of OxPL.

Which inflammasome is activated by POVPC, PGPC and LPC. The authors report no involvement of Casp-11 in oxPAPC-induced IL-1b release, but which inflammasome is involved then? NLRP3 involvement could easily be tested using MCC950.

Figure 1D/E, 2F, 3b, 3F, 4D-F: These blots require a loading control. It would be better to show combined supernatant and lysates samples, and blot for Casp-1, IL-1b, and a loading control (b-actin). (same for blots in Figure S1).

Figure 2C-F: It is unclear if dPGPC or dPOVPC means PGPCPOVPC sn-2 isolated after AOAH incubation or just PGPC/POVPC treated with AOAH. Please be more specific. If it is PGPC/POVPC treated with AOAH, then why do we not see a strong activation of the inflammasome due to the LPC that is produced?

Is AOAH still in the preps containing dPGPC etc..? If yes, a necessary control is to incubate cells with AOAH.

Figure 3F: this blot is hardly convincing. It appears that there is similar levels of IL-1b induced by LPC and dLPC.

Figure 4A: AOAH-/- cells appear to have a viability defect or grow more slowly, as they have lower crystal violet levels even without treatment. Thus all other values need to be normalized by setting the crystal violet values of untreated cells as 100%. I believe this will significantly change the data.

Figure 4E: please show LDH and Il-1b data as well, and quantify the blots as done for other blots. It looks like the actual data contradict the authors conclusions. Taking for example Figure 3F as an example (where the authors claim a significant difference), I would actually claim that there is more Casp-1 p20 even upon Dotap transfection of PGPC and POVPC.

Priming of AOAH expression: The authors mention that AOAH expression needs to be induced and that oxPL are poor inducers. This makes me wonder how AOAH can reduce cell death unprimed cells in Figure 4A-C for example. It would be helpful for the reader if the authors would provide blots showing AOAH expression in figure 4A-C.

Animal model: It is not evident from the manuscript why the authors chose the intranasal installation model. Please explain. Previous papers investigating the effects of oxPL used i.p. injections.

The authors should clearly point out that their data contradict findings by Zanoni et al., 2016, which proposed IL-1b release in dependence of Casp-11.

It would be helpful if the authors could include the chemical structures of OxPAPC, POVPC, LPC etc.., and how they are converted by AOAH incubation.

*Reviewer #3 (Recommendations for the authors):*

While the study is overall quite thorough, and of general interest, there are several clarifications, additional analyses, and additional data that could overall strengthen the findings. These are outlined below.

Generally there are some issues with the statistical analysis. It seems the Mann Whitney test was used to compare 2 groups, but in many of the experiments there are multiple groups, which indicates a need for using ANOVA tests. Also it is not always clear how many repeats were done, and the reasoning behind mega-analysis for some experiments, and representative samples for other experiments. The number of repeats for each experiment should be clarified and justified by the use of power analysis.

Figure 1:

D and E should be quantified as compared to their loading control. There is also do indication of percent of total input, or controls for differences when the supernatants are concentrated. Cleaved caspase-1 levels should also be examined in the cell lysate as well as the media.

Figure 2:

A and B do not appear to have statistics performed. Also where there is no indication of statistical significance, does this mean it wasn't analyzed or that it is not significant. This should be clarified. What statistics are performed should be described in the figure legends.

As with Figure 1 4 needs additional controls and quantification.

It says one representative experiment of 3 or 4 is shown, but it should be clarified which is 3 and which is 4. Also if possible a mega-analysis should be performed for optimal statistics. If this is not possible (due to a range of values between experiments) this should be specified. In order for a representative experiment to be shown all experiments must be statistically significant, or a mega-analysis needs to be done.

Figure 3:

This figure has many of the same concerns as the other figures in terms of quantification of the WB, and also how the statistics were done precisely. What the stars indicate should be indicated in every figure legend (not just M and Ms). It is unclear why this figure has a quantification of the WB, but other figures do not. It should be done on all, and also everything should be relative to the loading control and percent input.

Figure 4:

It seems like some of the experiments were only does twice. Did statistical analysis allow for this to be an acceptable number? This should be clarified.

Figure 5:

It is unclear why the quantification of cytokines and other inflammatory factors is done at the transcriptional level. Given the post-transcriptional regulation of many of these factors, and the availability of BALF, this data could be greatly strengthened by inclusion of some protein level data for key factors. Also some explanation as to why these specific cytokines, chemokines, and IRAK-M were chosen would be useful. What is the significance of IRAK-M in the context of acute lung injury for example?

It is interesting, that in contrast to the other models, in AOAH-/- mice LPS alone induced an increase in monocytes. This should be addressed in the discussion.

Figure 6:

It is encouraging that much of the transcriptional data recapitulates at the protein level, but this should also be shown in Figure 5.

Figure 7:

The histology in Figure 7 is a very important point, but for this to have any relevance it must be quantified and scored by a pathologist in a blinded fashion. Given that there is usually a wide range of phenotypes in different regions of the lung in this model the whole lung needs to be examined for proper quantification.

In addition, BAL protein and/or Evan's Blue assays should be done to determine how this damage compares to that seen in the other in vivo models. This is particular relevant as one would expect more damage in this model in the control animals.

---

## [Author Response]

Essential revisions:The manuscript demonstrates that endogenous oxidized phospholipids (OxPLs) can stimulate inflammatory responses in host cells, including induction of cell death and processing of interleukin-1 β. The manuscript futher identifies a host enzyme, acyloxyacyl hydrolase (AOAH), that cleaves oxidized phospholipids and thereby reduces their pro-inflammatory properties. A combination of in vitro and in vivo experiments supports the anti-inflammatory effects of AOAH. These are important findings that shed light on the mechanisms that limit damaging inflammation and we expect the findings to be of broad interest. The reviewers agreed that some technical aspects of the experiments and/or their presentation could be improved. We encourage submission of a revised version that address the following essential revisions:(1) One of the main claims of the manuscript is that OxPLs have pro-inflammatory effects that include inflammasome activation (independent of Casp11). This claim is somewhat at odds with at least some of the existing literature which claims that OxPLs inhibit Casp11 inflammasome activation and/or synergize with LPS to produce enhanced Casp11-dependent IL-1 release. The apparent discrepancies with the literature are not adequately discussed or addressed.(a) We require more discussion of how the new claims of this manuscript are to be reconciled with previous claims in the literature (e.g., see papers cited by R2).

We thank the reviewers for pointing out this important issue. We have added a paragraph in the Discussion and another in the Materials and methods.

Discussion

“Previous studies have shown that oxPLs may have both pro-inflammatory and anti-inflammatory bioactivities (Bochkov et al., 2002; Erridge et al., 2008). […] Whether AOAH also plays a role in regulating or promoting the anti-inflammatory activities of oxPLs awaits further investigation.”

Materials and methods

Prime and stimulate macrophages

“Many experimental variables may influence the impact that oxidized phospholipids have on cells in vitro, and both pro- and anti-inflammatory responses have been reported (Bochkov et al., 2002; Chu et al., 2018; Erridge et al., 2008; Mauerhofer et al., 2016; Muri et al., 2020; Oskolkova et al., 2010). […] The cells studied are also important, as some (macrophages and dendritic cells) produce AOAH and others (THP-1 cells, human endothelial cells) do not.”

(b) To support the main claim of inflammasome activation by OxPLs, we require that the authors address which inflammasome is being activated. At a minimum, we require that they test for the possible involvement of NLRP3 either with NLRP3 KO cells or MCC950 inhibitor.

Zanoni et al., found that caspase 11 can bind oxPAPC and that the NLRP3 inflammasome is required for oxPAPC-induced IL-1β release in primed dendritic cells (Zanoni et al., 2016). Later, the same group found that PGPC or POVPC but not oxPAPC can induce IL-1β release from primed bone marrow derived macrophages (BMDM) in a NLRP3-, Caspase 1/11-dependent manner (Zanoni et al., 2017). Yeon et al., also found that POVPC induced IL-1β and processed caspase 1 release from primed BMDM, which required NLRP3 (Yeon et al., 2017). In contrast, Muri et al., found that caspase 8 but not caspase 1 or NLRP3 was required for cyclo-epoxycyclopentenone-induced IL-1β release in primed bone marrow-derived dendritic cells or macrophages.

We found that NLRP3 specific inhibitor MCC950 reduced PGPC or LPC-induced inflammasome activation and IL-1β release. To our surprise, using inhibitors we found that in addition to caspase 1, caspase 8 was also indispensable, suggesting that caspase 8 may cleave caspase 1 and activated caspase 1 cleaves pro-IL-1β (Chi et al., 2014; Philip et al., 2014). Please see lines 95-105, new Figure 1E, F and new Figure 3B, C.

(2) We require that technical issues with many of the experiments be addressed:(a) All immunoblots require loading controls (see comments from all reviewers).

Thanks. We added actin controls to all blots. As 12.5 μg/ml oxPLs caused about 80% cell death of macrophages, it is hard to use cell actin as loading control for IL-1β or caspase 1 p20 in culture medium. As suggested by Reviewer 2, we used medium + cell mixture to blot for Caspase 1 (p20) and actin. When 5 μg/ml oxPLs were used, 80% of cells were alive; we used culture medium for caspase 1 (p20) and cell lysate for actin measurement. We have updated the Method, lines 394-403.

(b) Figure 2C-F: a better description of the experiment is required (see comments from R2).

dPGPC/dPOVPC means PGPC/POVPC treated with AOAH. AOAH can release both sn-2 and sn-1 fatty acyl chains from PGPC/POVPC. In addition, AOAH deacylates LPC. Please see Figure 2A, B and Figure 3A. We have clarified the definition of dPGPC/dPOVPC, line 120.

The samples were frozen after treatment. Freezing in the absence of glycerol inactivates AOAH. We added a sentence to make it clear, lines 420, 421.

(c) The blot in 3F needs quantification; if the effect is not significant, more caution is warranted.

We have omitted Figure 3F. After treating unprimed or primed AOAH WT and KO macrophages with lower concentration of LPC (5 μg/ml or 10 μM), we found that AOAH KO macrophages had greater caspase 1 cleavage than did WT macrophages, evidence that AOAH prevents LPC-induced inflammasome activation. Please see new Figure 3E.

(d) Clarify if AOAH KO cells have a growth defect, and if so, normalize the data in Figure 4 as requested by R2.

For old Figure 4A. Medium only (no treatment) groups, there is no significant difference in crystal violet OD between AOAH WT and KO macrophages. When we normalized KO macrophages ODs based on OD_medium_, we found that there were fewer adherent KO macrophages than WT macrophages when they were primed by Pam3 and treated with PGPC or POVPC, suggesting that AOAH prevents cell death in primed macrophages.

(e) Address the possible confounding issues associated with using Pam3CSK4 as a priming agent (see comments from R1).

As AOAH does not regulate acute responses to LPS, G+ bacteria, poly I:C (Lu et al., 2008) or Pam3 (Figure 4C, IL-6) in vitro or in vivo (Lu et al., 2008; Zou et al., 2017), we do not expect AOAH to modulate the priming effects of Pam3 or LPS.

To address this issue we tested CpG, which can also prime macrophages for oxPL-induced inflammasome activation and is not an AOAH substrate. We found that when AOAH WT and KO macrophages were primed with CpG, PGPC induced more cell death and IL-1β release from AOAH KO macrophages. Please see lines 154-159 and new Figure 4E.

(f) Address the issues raised about ZVAD/VX765 (see R1) and modify conclusions as necessary.

We have added inhibitor-only controls and found that those inhibitors did not regulate cell death or IL-1β release in our experimental settings (New Figure 1E, Figure 3B). It is interesting that both casp1 and casp8 inhibitors blocked IL-1β release and neither of them prevented cell death. We speculate that oxPLs may activate caspase 8 to cleave caspase 1, as found previously (Philip et al., 2014).

(g) Address the statistical issues identified by R3.

We have added statistical tests suggested by Reviewer 3. When multiple groups were compared, we used one-way ANOVA first and then we used Mann Whitney test for between- group comparisons. When *Aoah^+/+^* and *Aoah^-/-^* groups with multiple treatments were compared, we used two-way ANOVA first and then we used Mann Whitney test to compare the two groups with each specific treatment. We have added the number of repeats and the sample numbers for each experiment.

(3) The in vivo experiments using the HCl ventilation model (Figure 7) are particularly central to the main conclusion of the paper, namely, the claim that endogenous OxPLs are counteracted by AOAH (The other in vivo experiments involve provision of exogenous inflammatory lipids). As such, the data provided with this model in Figure 7 should be solidified:(a) Improve the analysis of the histological findings in Figure 7 as suggested by R3.(b) Quantify additional inflammatory markers, e.g., quantification of cytokines at the protein level, and markers of lung damage, e.g., quantification of Evans Blue and/or BAL protein as suggested by R3.

Thanks. We have scored the histology sections according to the lung injury scoring system recommended by the American Thoracic Society, measured BALF cytokines and BALF protein. Please see new Figure 7A, C and E.

Reviewer #1 (Recommendations for the authors):(1) It is difficult to know how to reconcile this work with other work in the field. It has been claimed that OxPLs antagonize Casp11 and are thus anti-inflammatory, whereas the current work suggests they are pro-inflammatory independent of Casp11. A more straightforward discussion of previous work and any relevant differences is necessary to help a reader put the present work in context.

We thank the reviewer for raising this important question. We think the oxPL species used and their concentrations, the routes of MAMP and oxPL delivery, and the order of addition of MAMP and oxPLs may contribute to the observations made in different laboratories. We have added a paragraph in the Discussion and another in the Methods, lines 298-325 and lines 347-358 (highlighted).

(2) If Casp11 is not activated by OxPLs, which inflammasome is? At a minimum a simple experiment with NLRP3 KO cells (or MCC950) could rule in or out NLRP3, which is the most obvious candidate.

We found that NLRP3 specific inhibitor MCC950 reduced PGPC or LPC-induced inflammasome activation and IL-1β release. To our surprise, using inhibitors we found that in addition to caspase 1, caspase 8 was also indispensable, suggesting that caspase 8 may cleave caspase 1 and activated caspase 1 cleaves pro-IL-1β (Chi et al., 2014; Philip et al., 2014). Please see lines 95-105, new Figure 1E, F and new Figure 3B, C.

(3) Although it is appreciated that priming is necessary for this system, it is potentially confouding that many experiments use a lipid (Pam3CSK4) to prime since AOAH may potentially be acting on this lipid (and this may in part explain some of the observed effects). Can the authors address this possibility a bit more cleanly? Why not prime with a cytokine (e.g., TNF?) that would not itself be a possible substrate of AOAH?

Thanks for the suggestion. Because AOAH does not regulate acute responses to LPS (Lu et al., 2008) or Pam3 (Figure 4C, IL-6) in vitro or in vivo (Lu et al., 2008; Zou et al., 2017), we do not expect AOAH to modulate the priming effects of Pam3 or LPS. To explore the possibility that inactivating the priming agent prevents inflammasome activation, we tested CpG, which can also prime macrophages for oxPL-induced inflammasome activation and is not an AOAH substrate. We found that when AOAH WT and KO macrophages were primed with CpG, PGPC induced more cell death and IL-1β release from AOAH KO macrophages than from WT macrophages. Please see lines 154-159 and new Figure 4E.

(4) The experiments with ZVAD in S1 lack a ZVAD only control. In my experience ZVAD itself is highly toxic to macropahges so this control seems critical. In addition, since there is a CASP8 pathway downstream of inflammasome activation, CASP1 inhibitors are not sufficient to reach the conclusion that cell death is inflammasome-independent (e.g., p.6).

Thanks for the good suggestions. We have added inhibitor-only controls and found that those inhibitors including ZVAD did not regulate cell death or IL-1β release in our experimental settings (New Figure 1E, Figure 3B). To our surprise, using inhibitors we found that in addition to caspase 1, caspase 8 was also indispensable, suggesting that caspase 8 may cleave caspase 1 and activated caspase 1 cleaves pro-IL-1β (Chi et al., 2014; Philip et al., 2014). Please see lines 95-105, new Figure 1E, F and new Figure 3B, C.

(5) Many of the western blots lack loading controls.

Thanks. We have added actin loading controls.

Reviewer #2 (Recommendations for the authors):In this manuscript Zou et al., investigate the role of the lipase AOAH on the pro-inflammatory roles of oxidized lipids (PGPC, POVPC). They show that the removal of sn-1 and sn-2 acyl chains from these lipids by AOAH dampens oxPL induced inflammasome activation, in vitro and in vivo. They further demonstrate that AOAH also dampens inflammasome activation evoqued by LPC. In conclusion, AOAH reduces tissue inflammation and cell death in response to endogenous oxidized lipids.Oxidized lipids have diverse biological activities, among these the induction of pathology and inflammation. Thus, the identification of an endogenous regulator that can dampen these effects is novel and interesting. The manuscript is understandable and the experiments can be easily followed. However some important controls are missing, and some point require clarification. Thus, the manuscript would improve if the following points would be addressed:The OxPL field is highly controversial. For example, OxPAPAC has also been shown to block non-canonical inflammasome activation (PMID: 29520027), and in general acts as an anti-inflammatory molecule (PMID: 32234476). The authors show discuss these discrepancies and ideally provide experimental data to show how AOAH affect the anti-inflammatory functions of OxPL.

Thanks for pointing out this important issue. We think the oxPL species used and their concentrations, the routes of MAMP and oxPL delivery, and the order of addition of MAMP and oxPLs may contribute to the observations made in different laboratories. We have added a paragraph in the Discussion and another in the Methods, lines 298-325 and lines 347-358.

Which inflammasome is activated by POVPC, PGPC and LPC. The authors report no involvement of Casp-11 in oxPAPC-induced IL-1b release, but which inflammasome is involved then? NLRP3 involvement could easily be tested using MCC950.

Thanks for the advice. Using NLRP3 specific inhibitor MCC950, caspase 1 inhibitor VX-765, and caspase 8 inhibitor Z-IETD-FMK, we found that NLRP3, caspase 1 and caspase 8 were all indispensable for PGPC or LPC-induced inflammasome activation. Please see new Figure 1E, F and Figure 3B, C.

Figure 1D/E, 2F, 3b, 3F, 4D-F: These blots require a loading control. It would be better to show combined supernatant and lysates samples, and blot for Casp-1, IL-1b, and a loading control (b-actin). (same for blots in Figure S1).

Thanks for the suggestion. We have added actin loading controls to all Western blots.

Figure 2C-F: It is unclear if dPGPC or dPOVPC means PGPCPOVPC sn-2 isolated after AOAH incubation or just PGPC/POVPC treated with AOAH. Please be more specific. If it is PGPC/POVPC treated with AOAH, then why do we not see a strong activation of the inflammasome due to the LPC that is produced?Is AOAH still in the preps containing dPGPC etc..? If yes, a necessary control is to incubate cells with AOAH.

AOAH removes the sn-2 moiety then the sn-1 palmitate, so little LPC remains (see Figure 2A, B – the released palmitate is shown). The reaction mixtures were quick-frozen prior to storage; AOAH is denatured by freezing in the absence of glycerol. We added a sentence to make it clear, lines 420, 421.

Figure 3F: this blot is hardly convincing. It appears that there is similar levels of IL-1b induced by LPC and dLPC.

We have omitted the old Figure 3F. After treating unprimed or primed AOAH WT and KO macrophages with lower concentration of LPC (5 μg/ml or 10 μM), we found that AOAH KO macrophages had greater caspase 1 cleavage than did WT macrophages, evidence that AOAH prevents LPC-induced inflammasome activation. Please see new Figure 3E.

Figure 4A: AOAH-/- cells appear to have a viability defect or grow more slowly, as they have lower crystal violet levels even without treatment. Thus all other values need to be normalized by setting the crystal violet values of untreated cells as 100%. I believe this will significantly change the data.

Thanks. For Figure 4A. Medium only (no treatment), there is no significant difference in crystal violet OD between AOAH WT and KO macrophages. After we normalized KO peritoneal macrophages ODs based on OD_medium_, we found that there were fewer adherent KO macrophages than WT macrophages when they were primed by Pam3 and treated with PGPC or POVPC, suggesting that AOAH prevents cell death in primed macrophages.

Figure 4E: please show LDH and Il-1b data as well, and quantify the blots as done for other blots. It looks like the actual data contradict the authors conclusions. Taking for example Figure 3F as an example (where the authors claim a significant difference), I would actually claim that there is more Casp-1 p20 even upon Dotap transfection of PGPC and POVPC.

We have repeated the experiment 4 more times with actin controls and found that when oxPLs were delivered by DOTAP to AOAH WT and KO macrophages, the macrophages had similar inflammasome activation and IL-1β release. Please see new Figure 4F, G. Cells did not die with the treatment of 5 μg/ml PGPC or POVPC encapsulated in DOTAP.

Priming of AOAH expression: The authors mention that AOAH expression needs to be induced and that oxPL are poor inducers. This makes me wonder how AOAH can reduce cell death unprimed cells in Figure 4A-C for example. It would be helpful for the reader if the authors would provide blots showing AOAH expression in figure 4A-C.

Our previous data revealed that resident peritoneal macrophages express quite high levels of AOAH and that LPS or Pam3 priming can slightly increase AOAH mRNA. Now we found that AOAH decreased oxPL-induced cell death in primed macrophages, suggesting that increased AOAH expression may help prevent oxPL-induced cell death.

Animal model: It is not evident from the manuscript why the authors chose the intranasal installation model. Please explain. Previous papers investigating the effects of oxPL used i.p. injections.

Imai et al., found that HCL/Ventilation and influenza virus infection induces oxPLs in the lung (Imai et al., 2008). We think intranasal instillation models are more clinically relevant than i.p. injection models. We have added a sentence to the Methods to be clear, lines 443-445.

The authors should clearly point out that their data contradict findings by Zanoni et al., 2016, which proposed IL-1b release in dependence of Casp-11.

Thanks. We have added a sentence to the paper to note this difference, lines 105-108.

It would be helpful if the authors could include the chemical structures of OxPAPC, POVPC, LPC etc.., and how they are converted by AOAH incubation.

Good point. We have added the structure diagrams to Figure 2, 3 and showed AOAH cleavage sites.

Reviewer #3 (Recommendations for the authors):While the study is overall quite thorough, and of general interest, there are several clarifications, additional analyses, and additional data that could overall strengthen the findings. These are outlined below.Generally there are some issues with the statistical analysis. It seems the Mann Whitney test was used to compare 2 groups, but in many of the experiments there are multiple groups, which indicates a need for using ANOVA tests. Also it is not always clear how many repeats were done, and the reasoning behind mega-analysis for some experiments, and representative samples for other experiments. The number of repeats for each experiment should be clarified and justified by the use of power analysis.

Thanks for pointing out the important statistical problems. When multiple groups were compared, we used one-way ANOVA first and then we used Mann Whitney test for between- group comparisons. When *Aoah^+/+^* and *Aoah^-/-^* groups with multiple treatments were compared, we used two-way ANOVA first and then we used Mann Whitney test to compare the two groups with each specific treatment. We have added the number of repeats and the sample numbers for each experiment.

Figure 1:D and E should be quantified as compared to their loading control. There is also do indication of percent of total input, or controls for differences when the supernatants are concentrated. Cleaved caspase-1 levels should also be examined in the cell lysate as well as the media.

As 12.5 μg/ml oxPLs caused 80% cell death in peritoneal macrophages, it is hard to use cell actin as the loading control for medium IL-1β or caspase 1 (p20). We combined medium + cell mixture to blot for caspase 1 (p20) and actin as suggested by Reviewer 2. All of the concentrated medium and cell mixture was loaded on the gel. When 5 μg/ml oxPLs were used and > 80% cells were alive, we used medium for caspase 1 (p20) and cells for actin detection. We added the detailed protocol to the Method, lines 394-403.

Figure 2:A and B do not appear to have statistics performed. Also where there is no indication of statistical significance, does this mean it wasn't analyzed or that it is not significant. This should be clarified. What statistics are performed should be described in the figure legends.As with Figure 1 4 needs additional controls and quantification.It says one representative experiment of 3 or 4 is shown, but it should be clarified which is 3 and which is 4. Also if possible a mega-analysis should be performed for optimal statistics. If this is not possible (due to a range of values between experiments) this should be specified. In order for a representative experiment to be shown all experiments must be statistically significant, or a mega-analysis needs to be done.

We have added statistical tests to A and B and increased the size of the graphs so that the error bars are more visible. We have added statistics to each figure legend.

We combined data of 3 experiment for Figure 2C-D and did mega-analysis.

Figure 3:This figure has many of the same concerns as the other figures in terms of quantification of the WB, and also how the statistics were done precisely. What the stars indicate should be indicated in every figure legend (not just M and Ms). It is unclear why this figure has a quantification of the WB, but other figures do not. It should be done on all, and also everything should be relative to the loading control and percent input.

Thanks. We added statistical methods to each legend and indicated what the stars mean. We have added loading controls to Western blots and quantitated each blot.

Figure 4:It seems like some of the experiments were only does twice. Did statistical analysis allow for this to be an acceptable number? This should be clarified.

In Figure 4, F and H each had 2 experiments and in each experiment, n = 3 and 5 respectively. In Figure 4F, we tested both PGPC and POVPC and the results were confirmed in Figure 4G.

Figure 5:It is unclear why the quantification of cytokines and other inflammatory factors is done at the transcriptional level. Given the post-transcriptional regulation of many of these factors, and the availability of BALF, this data could be greatly strengthened by inclusion of some protein level data for key factors. Also some explanation as to why these specific cytokines, chemokines, and IRAK-M were chosen would be useful. What is the significance of IRAK-M in the context of acute lung injury for example?It is interesting, that in contrast to the other models, in AOAH-/- mice LPS alone induced an increase in monocytes. This should be addressed in the discussion.

We have measured cytokines in BALF. Please see new Figure 5E.

IRAK-M, a negative regulator of the TLR signaling pathway, is usually upregulated when inflammation is stimulated. We have added a sentence to lines 195-197.

OxPLs alone induced more monocyte recruitment in AOAH KO mouse lungs. We have added a sentence to lines 191-193.

Figure 6:It is encouraging that much of the transcriptional data recapitulates at the protein level, but this should also be shown in Figure 5.

Thanks. We have added more statistical tests to Figure 6.

Figure 7:The histology in Figure 7 is a very important point, but for this to have any relevance it must be quantified and scored by a pathologist in a blinded fashion. Given that there is usually a wide range of phenotypes in different regions of the lung in this model the whole lung needs to be examined for proper quantification.

Thanks. We have scored lung inflammation and injury in 10 fields from each lung (Matute-Bello et al., 2011). There were 8 mice/group. Please see revised Methods, lines 511-515.

In addition, BAL protein and/or Evan's Blue assays should be done to determine how this damage compares to that seen in the other in vivo models. This is particular relevant as one would expect more damage in this model in the control animals.

Good point. We have measured BALF protein. HCl + ventilation-induced and HCl + oxPL-induced alveolar leakage were both less severe than that induced by LPS + oxPLs. We have added a sentence in lines 240, 241.

References

Chi, W., Li, F., Chen, H., Wang, Y., Zhu, Y., Yang, X., Zhu, J., Wu, F., Ouyang, H., Ge, J.*, et al.* (2014). Caspase-8 promotes NLRP1/NLRP3 inflammasome activation and IL-1beta production in acute glaucoma. Proceedings of the National Academy of Sciences of the United States of America *111*, 11181-11186.

Imai, Y., Kuba, K., Neely, G.G., Yaghubian-Malhami, R., Perkmann, T., van Loo, G., Ermolaeva, M., Veldhuizen, R., Leung, Y.H., Wang, H.*, et al.* (2008). Identification of oxidative stress and Toll-like receptor 4 signaling as a key pathway of acute lung injury. Cell *133*, 235-249.

Lu, M., Varley, A.W., Ohta, S., Hardwick, J., and Munford, R.S. (2008). Host inactivation of bacterial lipopolysaccharide prevents prolonged tolerance following gram-negative bacterial infection. Cell host and microbe *4*, 293-302.

Matute-Bello, G., Downey, G., Moore, B.B., Groshong, S.D., Matthay, M.A., Slutsky, A.S., Kuebler, W.M., and Acute Lung Injury in Animals Study, G. (2011). An official American Thoracic Society workshop report: features and measurements of experimental acute lung injury in animals. American journal of respiratory cell and molecular biology *44*, 725-738.

Philip, N.H., Dillon, C.P., Snyder, A.G., Fitzgerald, P., Wynosky-Dolfi, M.A., Zwack, E.E., Hu, B., Fitzgerald, L., Mauldin, E.A., Copenhaver, A.M.*, et al.* (2014). Caspase-8 mediates caspase-1 processing and innate immune defense in response to bacterial blockade of NF-kappaB and MAPK signaling. Proceedings of the National Academy of Sciences of the United States of America *111*, 7385-7390.

Yeon, S.H., Yang, G., Lee, H.E., and Lee, J.Y. (2017). Oxidized phosphatidylcholine induces the activation of NLRP3 inflammasome in macrophages. Journal of leukocyte biology *101*, 205-215.

Zanoni, I., Tan, Y., Di Gioia, M., Broggi, A., Ruan, J., Shi, J., Donado, C.A., Shao, F., Wu, H., Springstead, J.R.*, et al.* (2016). An endogenous caspase-11 ligand elicits interleukin-1 release from living dendritic cells. Science *352*, 1232-1236.

Zanoni, I., Tan, Y., Di Gioia, M., Springstead, J.R., and Kagan, J.C. (2017). By Capturing Inflammatory Lipids Released from Dying Cells, the Receptor CD14 Induces Inflammasome-Dependent Phagocyte Hyperactivation. Immunity *47*, 697-709 e693.

Zou, B., Jiang, W., Han, H., Li, J., Mao, W., Tang, Z., Yang, Q., Qian, G., Qian, J., Zeng, W.*, et al.* (2017). Acyloxyacyl hydrolase promotes the resolution of lipopolysaccharide-induced acute lung injury. PLoS pathogens *13*, e1006436.